# FlexConv: Continuous Kernel Convolutions with Differentiable Kernel Sizes

**David W. Romero**[*,1], **Robert-Jan Bruintjes**[*,2],
**Erik J. Bekkers**[3], **Jakub M. Tomczak**[1], **Mark Hoogendoorn**[1], **Jan C. van Gemert**[2]
[1] Vrije Universiteit Amsterdam    [2] Delft University of Technology    [3] University of Amsterdam
The Netherlands
d.w.romeroguzman@vu.nl, r.bruintjes@tudelft.nl

## Abstract

When designing Convolutional Neural Networks (CNNs), one must select the size of the convolutional kernels before training. Recent works show CNNs benefit from different kernel sizes at different layers, but exploring all possible combinations is unfeasible in practice. A more efficient approach is to learn the kernel size during training. However, existing works that learn the kernel size have a limited bandwidth. These approaches scale kernels by dilation, and thus the detail they can describe is limited. In this work, we propose *FlexConv*, a novel convolutional operation with which high bandwidth convolutional kernels of learnable kernel size can be learned at a fixed parameter cost. *FlexNets* model long-term dependencies without the use of pooling, achieve state-of-the-art performance on several sequential datasets, outperform recent works with learned kernel sizes, and are competitive with much deeper ResNets on image benchmark datasets. Additionally, FlexNets can be deployed at higher resolutions than those seen during training. To avoid aliasing, we propose a novel kernel parameterization with which the frequency of the kernels can be analytically controlled. Our novel kernel parameterization shows higher descriptive power and faster convergence speed than existing parameterizations. This leads to important improvements in classification accuracy.

## 1 Introduction

The kernel size of a convolutional layer defines the region from which features are computed, and is a crucial choice in their design. Commonly, small kernels (up to 7px) are used almost exclusively and are combined with pooling to model long term dependencies (Simonyan & Zisserman, 2014; Szegedy et al., 2015; He et al., 2016; Tan & Le, 2019). Recent works indicate, however, that CNNs benefit from using convolutional kernels (*i*) of varying size at different layers (Pintea et al., 2021; Tomen et al., 2021), and (*ii*) at the same resolution of the data (Peng et al., 2017; Cordonnier et al., 2019; Romero et al., 2021). Unfortunately, most CNNs represent convolutional kernels as tensors of discrete weights and their size must be fixed prior to training. This makes exploring different kernel sizes at different layers difficult and time-consuming due to (*i*) the large search space, and (*ii*) the large number of weights required to construct large kernels.

A more efficient way to tune different kernel sizes at different layers is to *learn* them during training. Existing methods define a *discrete* weighted set of basis functions, e.g., shifted Delta-Diracs (Fig. 2b, Dai et al. (2017)) or Gaussian functions (Fig. 2c, Jacobsen et al. (2016); Shelhamer et al. (2019); Pintea et al. (2021)). During training they learn dilation factors over the basis functions to increase the kernel size, which crucially limits the bandwidth of the resulting kernels.

In this work, we present the *Flexible Size Continuous Kernel Convolution* (FlexConv), a convolutional layer able to learn *high bandwidth* convolutional kernels of varying size during training (Fig. 1). Instead of using discrete weights, we provide a *continuous parameterization* of convolutional kernels via a small neural network (Romero et al., 2021). This parameterization allows us to model continuous functions of arbitrary size with a fixed number of parameters. By multiplying the response of the neural network with a Gaussian mask, the size of the kernel can be learned during training (Fig. 2a). This allows us to produce detailed kernels of small sizes (Fig. 3), and tune kernel sizes efficiently.

---

[*] Equal contribution.

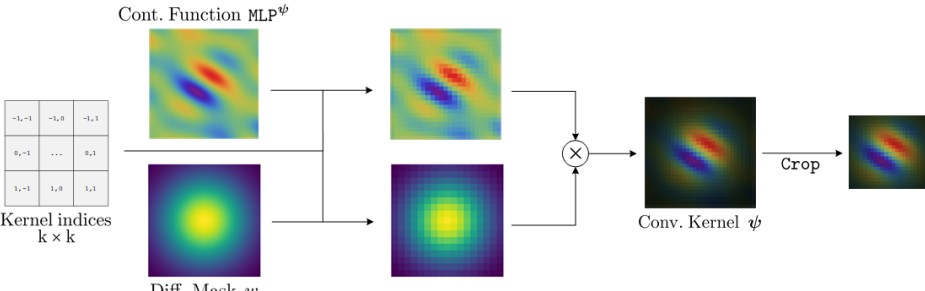

Figure 1: The Flexible Size Continuous Kernel Convolution (FlexConv). FlexConv defines convolutional kernels as the multiplication of a continuous convolutional kernel $\texttt{MLP}^{\psi}$, with a Gaussian mask of local support $w_{\text{gauss}}$: $\boldsymbol{\psi}(x,y) = w_{\text{gauss}}(x,y;\boldsymbol{\theta}_{\text{mask}}) \cdot \texttt{MLP}^{\psi}(x,y)$. By learning the parameters of the mask, the size of the convolutional kernel can be optimized during training. See also Fig. 7.

FlexConvs can be deployed at higher resolutions than those observed during training, simply by using a more densely sampled grid of kernel indices. However, the high bandwidth of the kernel can lead FlexConv to learn kernels that show aliasing at higher resolutions, if the kernel bandwidth exceeds the Nyquist frequency. To solve this problem, we propose to parameterize convolutional kernels as *Multiplicative Anisotropic Gabor Networks* (MAGNets). MAGNets are a new class of Multiplicative Filter Networks (Fathony et al., 2021) that allows us to analyze and control the frequency spectrum of the generated kernels. We use this analysis to regularize FlexConv against aliasing. With this regularization, FlexConvs can be directly deployed at higher resolutions with minimal accuracy loss. Furthermore, MAGNets provide higher descriptive power and faster convergence speed than existing continuous kernel parameterizations (Schütt et al., 2017; Finzi et al., 2020; Romero et al., 2021). This leads to important improvements in classification accuracy (Sec. 4).

Our experiments show that CNNs with FlexConvs, coined *FlexNets*, achieve state-of-the-art across several sequential datasets, match performance of recent works with learnable kernel sizes with less compute, and are competitive with much deeper ResNets (He et al., 2016) when applied on image benchmark datasets. Thanks to the ability of FlexConvs to generalize across resolutions, FlexNets can be efficiently trained at low-resolution to save compute, e.g., $16 \times 16$ CIFAR images, and be deployed on the original data resolution with marginal accuracy loss, e.g., $32 \times 32$ CIFAR images.

In summary, our **contributions** are:

- We introduce the *Flexible Size Continuous Kernel Convolution* (FlexConv), a convolution operation able to learn *high bandwidth* convolutional kernels of varying size end-to-end.

- Our proposed *Multiplicative Anisotropic Gabor Networks* (MAGNets) allow for analytic control of the properties of the generated kernels. This property allows us to construct analytic alias-free convolutional kernels that generalize to higher resolutions, and to train FlexNets at low resolution and deploy them at higher resolutions. Moreover, MAGNets show higher descriptive power and faster convergence speed than existing kernel parameterizations.

- CNN architectures with FlexConvs (FlexNets) obtain state-of-the-art across several sequential datasets, and match recent works with learnable kernel size on CIFAR-10 with less compute.

## 2 RELATED WORK

**Adaptive kernel sizes.** Loog & Lauze (2017) regularize the scale of convolutional kernels for filter learning. For image classification, adaptive kernel sizes have been proposed via learnable pixel-wise offsets (Dai et al., 2017), learnable padding operations (Han et al., 2018), learnable dilated Gaussian functions (Shelhamer et al., 2019; Xiong et al., 2020; Tabernik et al., 2020; Nguyen, 2020) and scalable Gaussian derivative filters (Pintea et al., 2021; Tomen et al., 2021; Lindeberg, 2021). These approaches either dilate discrete kernels (Fig. 2b), or use discrete weights on dilated basis functions (Fig. 2c). Using dilation crucially limits the bandwidth of the resulting kernels. In contrast, FlexConvs are able to construct high bandwidth convolutional kernels of varying size with a fixed parameter count. Larger kernels are obtained simply by passing more positions to the kernel network (Fig. 1).

**Continuous kernel convolutions.** Discrete convolutional kernel parameterizations assign an independent weight to each specific position in the kernel. Continuous convolutional kernels, on the

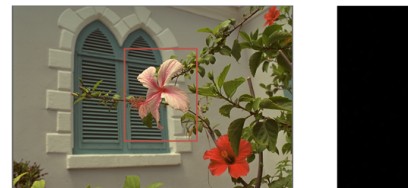 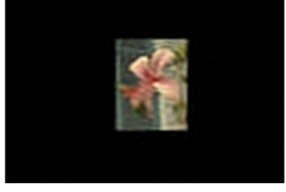 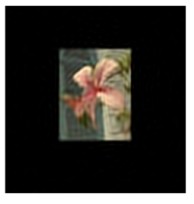 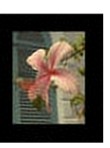 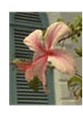

(a) FlexConv kernels (ours)     (b) Dilation / deformation (Dai et al., 2017)     (c) (Learnable) parametric dilation (Pintea et al., 2021)

Figure 2: Existing approaches increase the size of convolutional kernels via (learnable) parametric dilations, e.g., by deformation (b) or by Gaussian blur (c). However, dilation limits the bandwidth of the dilated kernel and with it, the amount of detail it can describe. Contrarily, FlexNets extend their kernels by passing a larger vector of positions to the neural network parameterizing them. As a result, FlexConvs are able to learn *high bandwidth* convolutional kernels of varying size end-to-end (a).

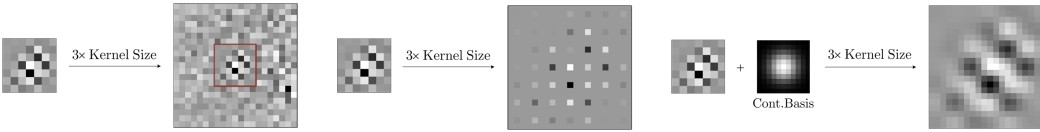

(a) Ground Truth       (b) Reconstructions at varying degrees of localization

Figure 3: The importance of dynamic sizes in continuous kernel convolutions. Consider a neural network predicting pixel values at each position. If the entire image is considered, the network must use part of its capacity to learn to predict zeros outside of the flower region, which in turn degrades the quality of the approximation in the region of interest (b). Importantly, the better the localization of the flower, the higher the approximation fidelity becomes. FlexNets learn the size of their convolutional kernels at each layer during training, and thus *(i)* use the capacity of the kernel efficiently, *(ii)* converge faster to good approximations, and *(iii)* are faster in execution –via dynamic cropping–.

other hand, view convolutional kernels as continuous functions parameterized via a small neural network $\texttt{MLP}^{\psi}: \mathbb{R}^D \to \mathbb{R}^{N_{out} \times N_{in}}$, with D the data dimensionality. This defines a convolutional kernel for which arbitrary input positions can be queried. Continuous kernels have primarily been used to handle irregularly-sampled data *locally*, e.g., molecular data (Simonovsky & Komodakis, 2017; Schütt et al., 2017) and point-clouds (Thomas et al., 2018; Wang et al., 2018; Shi et al., 2019).

Recently, Romero et al. (2021) introduced the Continuous Kernel Convolution (CKConv) as a tool to model long-term dependencies. CKConv uses a continuous kernel parameterization to construct convolutional kernels as big as the input signal with a constant parameter cost. Contrarily, FlexConvs jointly learn the convolutional kernel as well as its size. This leads to important advantages in terms of expressivity (Fig. 3), convergence speed and compute costs of the operation.

**Implicit neural representations.** Parameterizing a convolutional kernel via a neural network can be seen as learning an implicit neural representation of the underlying convolutional kernel (Romero et al., 2021). Implicit neural representations construct continuous data representations by encoding data in the weights of a neural network (Park et al., 2019; Sitzmann et al., 2020; Fathony et al., 2021).

We replace the SIREN (Sitzmann et al., 2020) kernel parameterization used in Romero et al. (2021) by our *Multiplicative Anisotropic Gabor Networks*: a new class of Multiplicative Filter Networks (Fathony et al., 2021). MFNs allow for analytic control of the resulting representations, and allow us to construct analytic alias-free convolutional kernels. The higher expressivity and convergence speed of MAGNets lead to accuracy improvements in CNNs using them as kernel parameterization.

## 3 METHOD

In this section, we introduce our approach. First, we introduce FlexConv and the Gaussian mask. Next, we introduce our Multiplicative Anisotropic Gabor Networks (MAGNets) and provide a description of our regularization technique used to control the spectral components of the generated kernel.

### 3.1 FLEXIBLE SIZE CONTINUOUS KERNEL CONVOLUTION (FLEXCONV)

To learn the kernel size during training, FlexConvs define their convolutional kernels $\psi$ as the product of the output of a neural network $\texttt{MLP}^{\psi}$ with a Gaussian mask of local support. The neural network $\texttt{MLP}^{\psi}$ parameterizes the kernel, and the Gaussian mask parameterizes its size (Fig. 1).

**Anisotropic Gaussian mask.** Let $G(x; \mu_X, \sigma_X^2) := \exp\left\{-\frac{1}{2}\sigma_X^{-2}(x-\mu_X)^2\right\}$ be a Gaussian function parameterized by a mean-variance tuple $(\mu_X, \sigma_X^2)$. The anisotropic Gaussian mask is defined as:

$$w_{\text{gauss}}(x, y; \{\mu_X, \sigma_X^2, \mu_Y, \sigma_Y^2\}) = G(x; \mu_X, \sigma_X^2)G(y; \mu_Y, \sigma_Y^2). \tag{1}$$

By learning $(\mu_X, \sigma_X^2)$ and $(\mu_Y, \sigma_Y^2)$ independently, anisotropic non-centered windows can be learned.

## 3.2 MULTIPLICATIVE ANISOTROPIC GABOR NETWORKS (MAGNETS)

In this section, we formalize our proposed parameterization for the kernel $\texttt{MLP}^\psi$. We start by introducing Multiplicative Filter Networks (Fathony et al., 2021), and present our MAGNets next.

**Multiplicative Filter Networks (MFNs).** Recently, Fathony et al. (2021) proposed to construct implicit neural representations as the linear combination of exponentially many basis functions $\mathbf{g}$:

$$\mathbf{h}^{(1)} = \mathbf{g}\left([x, y]; \boldsymbol{\theta}^{(1)}\right) \qquad\qquad \mathbf{g} : \mathbb{R}^2 \to \mathbb{R}^{N_{\text{hid}}} \tag{2}$$

$$\mathbf{h}^{(l)} = \left(\mathbf{W}^{(l)}\mathbf{h}^{(l-1)} + \mathbf{b}^{(l)}\right) \cdot \mathbf{g}\left([x, y]; \boldsymbol{\theta}^{(l)}\right) \qquad \mathbf{W}^{(l)} \in \mathbb{R}^{N_{\text{hid}} \times N_{\text{hid}}}, \mathbf{b}^{(l)} \in \mathbb{R}^{N_{\text{hid}}} \tag{3}$$

$$\psi(x, y) = \mathbf{W}^{(L)}\mathbf{h}^{(L-1)} + \mathbf{b}^{(L)} \qquad\qquad \mathbf{W}^{(L)} \in \mathbb{R}^{N \times N_{\text{hid}}}, \mathbf{b}^{(L)} \in \mathbb{R}^{N} \tag{4}$$

where $\left\{\boldsymbol{\theta}^{(l)}, \mathbf{W}^{(l)}, \mathbf{b}^{(l)}\right\}$ depict the learnable parameters of the bases and the affine transformations, and $N, N_{\text{hid}}$ depict the number of output and hidden channels, respectively. Depending on the selection of $\mathbf{g}$, MFNs obtain approximations comparable to those of SIRENs (Sitzmann et al., 2020) with faster convergence rate. The most successful instantiation of MNFs are the *Multiplicative Gabor Network* (MGN): MFNs constructed with isotropic Gabor functions as basis $\mathbf{g}$ (in Eq. 2):

$$\mathbf{g}\left([x, y]; \boldsymbol{\theta}^{(l)}\right) = \exp\left(-\frac{\boldsymbol{\gamma}^{(l)}}{2}\left[\left(x - \boldsymbol{\mu}^{(l)}\right)^2 + \left(y - \boldsymbol{\mu}^{(l)}\right)^2\right]\right)\text{Sin}\left(\mathbf{W}_g^{(l)} \cdot [x, y] + \mathbf{b}_g^{(l)}\right), \tag{5}$$

$$\boldsymbol{\theta}^{(l)} = \left\{\boldsymbol{\gamma}^{(l)} \in \mathbb{R}^{N_{\text{hid}}}, \boldsymbol{\mu}^{(l)} \in \mathbb{R}^{N_{\text{hid}}}, \mathbf{W}_g^{(l)} \in \mathbb{R}^{N_{\text{hid}} \times 2}, \mathbf{b}_g^{(l)} \in \mathbb{R}^{N_{\text{hid}}}\right\}. \tag{6}$$

Note that, by setting $N = N_{\text{out}} \times N_{\text{in}}$, an MFN can parameterize a convolutional kernel with $N_{\text{in}}$ input and $N_{\text{out}}$ output channels. Fathony et al. (2021) show that MFNs are equivalent to a linear combination of exponentially many basis functions $\mathbf{g}$. This allows us to analytically derive properties of MFN representations, and plays a crucial role in the derivation of alias-free MAGNets (Sec. 3.3).

**Multiplicative Anisotropic Gabor Networks (MAGNets).** Our MAGNet formulation is based on the observation that isotropic Gabor functions, i.e., with equal $\gamma$ for the horizontal and vertical directions, are undesirable as basis for the construction of MFNs. Whenever a frequency is required along a certain direction, an isotropic Gabor function automatically introduces that frequency in both directions. As a result, other bases must counteract this frequency in the direction where the frequency is not required, and thus the capacity of the MFN is not used optimally (Daugman, 1988).

Following the original formulation of the 2D Gabor functions (Daugman, 1988), we alleviate this limitation by using anisotropic Gabor functions instead:

$$\mathbf{g}\left([x, y]; \boldsymbol{\theta}^{(l)}\right) = \exp\left(-\frac{1}{2}\left[\left(\boldsymbol{\gamma}_X^{(l)}\left(x - \boldsymbol{\mu}_X^{(l)}\right)\right)^2 + \left(\boldsymbol{\gamma}_Y^{(l)}\left(y - \boldsymbol{\mu}_Y^{(l)}\right)\right)^2\right]\right)\text{Sin}\left(\mathbf{W}_g^{(l)}[x, y] + \mathbf{b}_g^{(l)}\right) \tag{7}$$

$$\boldsymbol{\theta}^{(l)} = \left\{\boldsymbol{\gamma}_X^{(l)} \in \mathbb{R}^{N_{\text{hid}}}, \boldsymbol{\gamma}_Y^{(l)} \in \mathbb{R}^{N_{\text{hid}}}, \boldsymbol{\mu}_X^{(l)} \in \mathbb{R}^{N_{\text{hid}}}, \boldsymbol{\mu}_Y^{(l)} \in \mathbb{R}^{N_{\text{hid}}}, \mathbf{W}_g^{(l)} \in \mathbb{R}^{N_{\text{hid}} \times 2}, \mathbf{b}_g^{(l)} \in \mathbb{R}^{N_{\text{hid}}}\right\}. \tag{8}$$

The resulting *Multiplicative Anisotropic Gabor Network* (MAGNet) obtains better control upon frequency components introduced to the approximation, and demonstrates important improvements in terms of descriptive power and convergence speed (Sec. 4).

**MAGNet initialization.** Fathony et al. (2021) proposes to initialize MGNs by drawing the size of the Gaussian envelopes, i.e., the $\boldsymbol{\gamma}^{(l)}$ term, from a $\text{Gamma}(\alpha \cdot L^{-1}, \beta)$ distribution at every layer $l \in [1, .., L-1]$. We observe however that this initialization does not provide much variability on the initial extension of the Gaussian envelopes and in fact, most of them cover a large portion of the space at initialization. To stimulate diversity, we initialize the $\{\boldsymbol{\gamma}_X^{(l)}, \boldsymbol{\gamma}_Y^{(l)}\}$ terms by a $\text{Gamma}(\alpha l^{-1}, \beta)$ distribution at the $l$-th layer. We observe that our proposed initialization consistently leads to better accuracy than the initialization of Fathony et al. (2021) across all tasks considered. (Sec. 4).

## 3.3 ANALYTIC ALIAS-FREE MAGNETS

FlexConvs can be deployed at higher resolutions than those observed during training, simply by sampling the underlying continuous representation of the kernel more densely, and accounting for the

change in sampling rate. Consider a D-dimensional input signal $f_{\mathrm{r}^{(1)}}$ with resolution $\mathrm{r}^{(1)}$. FlexConv learns a kernel $\psi_{\mathrm{r}^{(1)}}$ that can be inferred at a higher resolution $\mathrm{r}^{(2)}$ (Romero et al., 2021):

$$\left(f_{\mathrm{r}^{(2)}} * \psi_{\mathrm{r}^{(2)}}\right) \approx \left(\frac{\mathrm{r}^{(1)}}{\mathrm{r}^{(2)}}\right)^{\mathrm{D}} \left(f_{\mathrm{r}^{(1)}} * \psi_{\mathrm{r}^{(1)}}\right). \tag{9}$$

Note however, that Eq. 9 holds *approximately*. This is due to aliasing artifacts which can appear if the frequencies in the learned kernel surpass the Nyquist criterion of the target resolution. Consequently, an anti-aliased parameterization is vital to construct kernels that generalize well to high resolutions.

**Towards alias-free implicit neural representations.** We observe that SIRENs as well as unconstrained MFNs and MAGNets exhibit aliasing when deployed on resolutions higher than the training resolution, which hurts performance of the model. An example kernel with aliasing is shown in Fig. 8.

To combat aliasing, we would like to control the representation learned by MAGNets. MAGNets –and MFNs in general– construct implicit neural representations that can be seen as a *linear combination of basis functions*. This property allows us to analytically derive and study the properties of the resulting neural representation. Here, we use this property to derive the maximum frequency of MAGNet-generated kernels, so as to regularize MAGNets against aliasing during training. We analytically derive the maximum frequency of a MAGNet, and penalize it whenever it exceeds the Nyquist frequency of the training resolution. We note that analytic derivations are difficult for other implicit neural representations, e.g., SIRENs, due to stacked layer-wise nonlinearities.

**Maximum frequency of MAGNets.** The maximum frequency component of a MAGNet is given by:

$$f_{\mathrm{MAGNet}}^+ = \sum_{l=1}^{\mathrm{L}} \max_{i_l} \left( \left( \max_j \frac{\mathbf{W}_{\mathrm{g},i_l,j}^{(l)}}{2\pi} \right) + \frac{\sigma_{\mathrm{cut}} \min\{\boldsymbol{\gamma}_{\mathrm{X},i_l}^{(l)}, \boldsymbol{\gamma}_{\mathrm{Y},i_l}^{(l)}\}}{2\pi} \right), \tag{21}$$

where L corresponds to the number of layers, $\mathbf{W}_{\mathrm{g}}^{(l)}, \boldsymbol{\gamma}_{\mathrm{X}}^{(l)}, \boldsymbol{\gamma}_{\mathrm{Y}}^{(l)}$ to the MAGNet parameters as defined in Eq. 8, and $\sigma_{\mathrm{cut}} = 2 \cdot$ `stdev` to the cut-off frequency of the Gaussian envelopes in the Gabor filters. A formal treatment as well as the derivations can be found in Appx. A.1.

**Effect of the FlexConv mask.** The Gaussian mask used to localize the response of the MAGNet also has an effect on the frequency spectrum. Hence, the maximum frequency of a FlexConv kernel is:

$$f_{\mathrm{FlexConv}}^+ = f_{\mathrm{MAGNet}}^+ + f_{w_{\mathrm{gauss}}}^+, \quad \text{with} \;\; f_{w_{\mathrm{gauss}}}^+ = \frac{\sigma_{\mathrm{cut}}}{\max\{\sigma_{\mathrm{X}}, \sigma_{\mathrm{Y}}\} 2\pi}. \tag{22}$$

Here, $\sigma_{\mathrm{X}}, \sigma_{\mathrm{Y}}$ correspond to the mask parameters (Eq. 1). Intuitively, multiplication with the mask blurs in the frequency domain, as it is equivalent to convolution with the Fourier transform of the mask.

**Aliasing regularization of FlexConv kernels.** With the analytic derivation of $f_{\mathrm{FlexConv}}^+$ we penalize the generated kernels to have frequencies smaller or equal to their Nyquist frequency $f_{\mathrm{Nyq}}(k)$ via:

$$\mathcal{L}_{\mathrm{HF}} = \| \max\{f_{\mathrm{FlexConv}}^+, f_{\mathrm{Nyq}}(k)\} - f_{\mathrm{Nyq}}(k) \|^2, \quad \text{with} \;\; f_{\mathrm{Nyq}}(k) = \frac{k-1}{4}. \tag{25}$$

Here, $k$ depicts the size of the FlexConv kernel before applying the Gaussian mask, and is equal to the size of the input signal. In practice, we implement Eq. 25 by regularizing the individual MAGNet layers, as is detailed in Appx. A.2. To verify our method, Fig. 8 (Appx. A.1) shows that the frequency components of FlexNet kernels are properly regularized for aliasing.

## 4 EXPERIMENTS

We evaluate FlexConv across classification tasks on sequential and image benchmark datasets, and validate the ability of MAGNets to approximate complex functions. A complete description of the datasets used is given in Appx. B. Appx. D.2 reports the parameters used in all our experiments.[1]

### 4.1 WHAT KIND OF FUNCTIONS CAN MAGNETS APPROXIMATE?

**Bandwidth of methods with learnable sizes.** First, we compare the bandwidth of MAGNet against N-Jet (Pintea et al., 2021) by optimizing each to fit simple targets: (i) Gabor filters of known frequency, (ii) random noise and (iii) an a $11 \times 11$ AlexNet kernel from the first layer (Krizhevsky et al., 2012). Fig. 4 shows that, even with 9 orders of Gaussian derivatives, N-Jets cannot fit high frequency signals in large kernels. Crucially, N-Jet models require many Gaussian derivative orders to model high frequency signals in large kernels: a hyperparameter which proportionally increases their inference

---

[1]Our code is publicly available at `https://github.com/rjbruin/flexconv`.

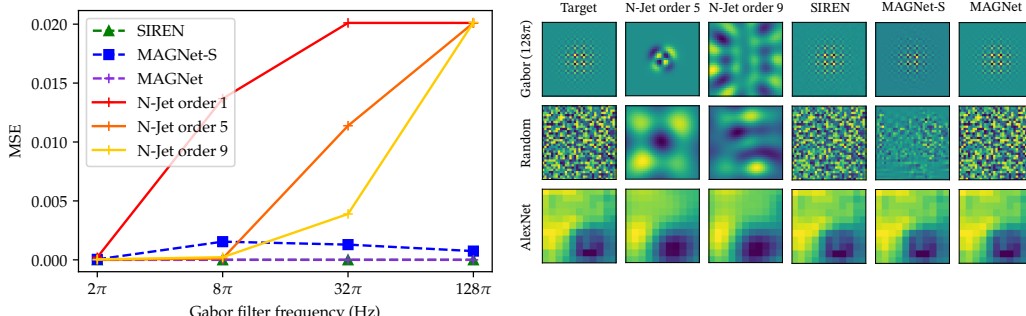

Figure 4: *Left*: Final MSE after fitting each model to Gabor filters of different frequencies. N-Jets cannot fit high frequencies. *Right*: Kernels learned by each model. SIREN and MAGNet can fit all targets. MAGNet-S: a small MAGNet of size akin to N-Jets, still does well on the Gabor and AlexNet targets.

Table 1: Test accuracy and ablation studies on sMNIST, pMNIST, sCIFAR10 and npCIFAR10.

| MODEL | SIZE | sMNIST | pMNIST | sCIFAR10 | npCIFAR10 |
|---|---|---|---|---|---|
| DilRNN (Chang et al., 2017) | 44K | 98.0 | 96.1 | - | - |
| IndRNN (Li et al., 2018) | 83K | 99.0 | 96.0 | - | - |
| TCN (Bai et al., 2018a) | 70K | 99.0 | 97.2 | - | - |
| r-LSTM (Trinh et al., 2018) | 0.5M | 98.4 | 95.2 | 72.2 | - |
| Self-Att. (Trinh et al., 2018) | 0.5M | 98.9 | 97.9 | 62.2 | - |
| TrellisNet (Bai et al., 2018b) | 8M | 99.20 | 98.13 | 73.42 | - |
| URLSTM (Gu et al., 2020b) | - | 99.28 | 96.96 | 71.00 | - |
| URGRU + Zoneout (Gu et al., 2020b) | - | 99.27 | 96.51 | **74.40** | - |
| HiPPO (Gu et al., 2020a) | 0.5M | - | **98.30** | - | - |
| Lipschitz RNN (Erichson et al., 2020) | 158K | 99.4 | 97.3 | 64.2 | 59.0 |
| coRNN (Rusch & Mishra, 2020) | 134K | **99.4** | 97.3 | - | 59.0 |
| UnICORNN (Rusch & Mishra, 2021) | 135K | - | 98.4 | - | **62.4** |
| pLMU (Chilkuri & Eliasmith, 2021) | 165K | - | 98.49 | - | - |
| CKCNN-2 | 98K | 99.31 | 98.00 | 62.25 | 60.5 |
| CKCNN-2-Big | 1M | 99.32 | 98.54 | 63.74 | 62.2 |
| CKTCN$_{FOURIER}$-2 | 105K | 99.44 | 98.40 | 68.28 | 66.26 |
| CKTCN$_{GABOR}$-2 | 106K | 99.52 | 98.38 | 69.26 | 67.37 |
| CKTCN$_{MAGNET}$-2 | 105K | **99.55** | **98.57** | **74.58** | **67.52** |
| FlexTCN-2 | 108K | **99.60** | 98.61 | 78.99 | 67.11 |
| FlexTCN-4 | 241K | **99.60** | 98.72 | 80.26 | 67.42 |
| FlexTCN-6 | 375K | **99.62** | 98.63 | 80.82 | 69.87 |
| FlexTCN$_{SIREN}$-6 | 343K | 99.03 | 95.36 | 69.24 | 57.27 |
| FlexTCN$_{Fourier}$-6 | 370K | 99.49 | 97.97 | 74.79 | 67.35 |
| FlexTCN$_{Gabor}$-6 | 373K | 99.50 | 98.37 | 78.36 | 67.56 |
| FlexTCN$_{MAGNet}$-6 | 375K | **99.62** | **98.63** | **80.82** | **69.87** |

time and parameter count. MAGNets, on the other hand, accurately model large high frequency signals. This allows FlexNets to learn large kernels with high frequency components.

**Expressivity of `MLP` parameterizations.** Next, we compare the descriptive power and convergence speed of MAGNets, Gabor MFNs, Fourier MFNs and SIRENs for image approximation. To this end, we fit the images in the Kodak dataset (Kodak, 1991) with each of these methods. Our results (Tab. 5) show that MAGNets outperform all other methods, and converge faster to good approximations.

## 4.2 CLASSIFICATION TASKS

**Network specifications.** Here, we specify our networks for all our classification experiments. We parameterize all our convolutional kernels as the superposition of a 3-layer MAGNet and a learnable anisotropic Gaussian mask. We construct two network instances for sequential and image datasets respectively: FlexTCNs and FlexNets. Both are constructed by taking the structure of a baseline network –TCN (Bai et al., 2018a) or CIFARResNet (He et al., 2016)–, removing all internal pooling layers, and replacing convolutional kernels by FlexConvs. The FlexNet architecture is shown in Fig. 10 and varies only in the number of channels and blocks, e.g., FlexNet-16 has 7 blocks. Akin to Romero et al. (2021) we utilize the Fourier theorem to speed up convolutions with large kernels.

**Mask initialization.** We initialize the FlexConv masks to be small. Preliminary experiments show this leads to better performance, faster execution, and faster training convergence. For sequences, the mask center is initialized at the last kernel position to prioritize the last information seen.

**Time series and sequential data.** First we evaluate FlexTCNs on sequential classification datasets, for which long-term dependencies play an important role. We validate our approach on intrinsic

Table 2: Test accuracy on CT, SC and SC_raw

| MODEL | SIZE | CT | SC | SC_RAW |
|---|---|---|---|---|
| GRU-ODE | 89K | 96.2 | 44.8 | ~10.0 |
| GRU-$\Delta t$ | 89K | 97.8 | 20.0 | ~10.0 |
| GRU-D | 89K | 95.9 | 23.9 | ~10.0 |
| ODE-RNN | 89K | 97.1 | 93.2 | ~10.0 |
| NCDE | 89K | 98.8 | 88.5 | ~10.0 |
| CKCNN | 100K | **99.53** | 95.27 | 71.66 |
| CKTCN$_{Fourier}$ | - | | 95.65 | 74.90 |
| CKTCN$_{Gabor}$ | - | | 96.66 | 78.10 |
| CKTCN$_{MAGNet}$ | 105K | **99.53** | **97.01** | **80.69** |
| FlexTCN-2 | 105sK | **99.53** | 97.10 | **88.03** |
| FlexTCN-4 | 239K | **99.53** | **97.73** | **90.45** |
| FlexTCN-6 | 373K | **99.53** | 97.67 | **91.73** |
| FlexTCN$_{SIREN}$-6 | 370K | - | 95.83 | 85.73 |
| FlexTCN$_{Fourier}$-6 | 342K | - | 97.62 | 91.02 |
| FlexTCN$_{Gabor}$-6 | 373K | - | 97.35 | 91.50 |
| FlexTCN$_{MAGNet}$-6 | 373K | - | **97.67** | **91.73** |

Table 3: Results on CIFAR-10. Results from *original works and † single run.

| MODEL | SIZE | CIFAR-10 ACC. | TIME (SEC/EPOCH) |
|---|---|---|---|
| CIFARResNet-44 | 0.66M | 92.9*† | 22 |
| DCN-$\sigma^{ji}$ | 0.47M | 89.7 ± 0.3* | - |
| N-Jet-CIFARResNet32 | 0.52M | 92.3 ± 0.3* | - |
| N-Jet-ALLCNN | 1.07M | 92.5 ± 0.1* | - |
| FlexNet-16 w/ conv. ($k = 3$) | 0.17M | 89.5 ± 0.3 | 41 |
| FlexNet-16 w/ conv. ($k = 33$) | 20.0M | 78.0 ± 0.3 | 242 |
| FlexNet-16 w/ N-Jet | 0.70M | 91.7 ± 0.1 | 409 |
| CKCNN-16 | 0.63M | 72.1 ± 0.2 | 68 |
| CKCNN$_{MAGNet}$-16 | 0.67M | 86.8 ± 0.6 | 102 |
| FlexNet$_{SIREN}$-16 | 0.63M | 89.0 ± 0.3 | 89 |
| FlexNet$_{Gabor}$-16 | 0.67M | 91.9 ± 0.2 | 161 |
| FlexNet$_{Gabor}$-16 + anis. Gauss. | 0.67M | 92.0 ± 0.1 | 147 |
| FlexNet$_{Gabor}$-16 + Gabor init. | 0.67M | 92.0 ± 0.2 | 150 |
| FlexNet-16 | 0.67M | 92.2 ± 0.1 | 127 |

discrete data: *sequential MNIST*, *permuted MNIST* (Le et al., 2015), *sequential CIFAR10* (Chang et al., 2017), *noise-padded CIFAR10* (Chang et al., 2019), as well as time-series data: *CharacterTrajectories* (CT) (Bagnall et al., 2018), *SpeechCommands* (Warden, 2018) with raw waveform (SC_raw) and MFCC input representations (SC).

Our results are summarized in Tables 1 and 2. FlexTCNs with two residual blocks obtain state-of-the-art results on all tasks considered. In addition, depth further improves performance. FlexTCN-6 improves the current state-of-the-art on sCIFAR10 and npCIFAR10 by more than 6%. On the difficult SC_raw dataset –with sequences of length 16000–, FlexTCN-6 outperform the previous state-of-the-art by 20.07%: a remarkable improvement.

Furthermore, we conduct ablation studies by changing the parameterization of $\texttt{MLP}^\psi$, and switching off the learnable kernel size ("CKTCNs") and considering global kernel sizes instead. CKTCNs and FlexTCNs with MAGNet kernels outperform corresponding models with all other kernel parameterizations: SIRENs (Sitzmann et al., 2020), MGNs and MFNs (Fathony et al., 2021). Moreover, we see a consistent improvement with respect to CKCNNs (Romero et al., 2021) by using learnable kernel sizes. This shows that both MAGNets and learnable kernel sizes contribute to the performance of FlexTCNs. Note that in 1D, MAGNets are equivalent to MGNs. However, MAGNets consistently perform better than MGNs. This improvement in accuracy is a result of our MAGNet initialization.

**Image classification.** Next, we evaluate FlexNets for image classification on CIFAR-10 (Krizhevsky et al., 2009). Additional experiments on Imagenet-32, MNIST and STL-10 can be found in Appx. C.

Table 3 shows our results on CIFAR-10. FlexNets are competitive with pooling-based methods such as CIFARResNet (He et al., 2016) and outperform learnable kernel size method DCNs (Tomen et al., 2021). In addition, we compare using N-Jet layers of order three (as in Pintea et al. (2021)) in FlexNets against using MAGNet kernels. We observe that N-Jet layers lead to worse performance, and are significantly slower than FlexConv layers with MAGNet kernels. The low accuracy of N-Jet layers is likely to be linked to the fact that FlexNets do not use pooling. Consequently, N-Jets are forced to learn large kernels with high-frequencies, which we show N-Jets struggle learning in Sec. 4.1.

To illustrate the effect of learning kernel sizes, we also compare FlexNets against FlexNets with large and small discrete convolutional kernels (Tab. 3). Using small kernel sizes is parameter efficient, but is not competitive with FlexNets. Large discrete kernels on the other hand require a copious amount of parameters and lead to significantly worse performance. These results indicate that the best solution is somewhere in the middle and varying kernel sizes can learn the optimal kernel size for the task at hand.

Similar to the sequential case, we conduct ablation studies on image data with learnable, non-learnable kernel sizes and different kernel parameterizations. Table 3 shows that FlexNets outperform CKCNNs with corresponding kernel parameterizations. In addition, a clear difference in performance is apparent for MAGNets with respect to other parameterizations. These results corroborate that both MAGNets and FlexConvs contribute to the performance of FlexNets. Moreover, Tab. 3 illustrates the effect of the two contributions of MAGNet over MGN: anisotropic Gabor filters, and our improved initialization. Our results in image data are in unison with our previous results for sequential data (Tabs. 1, 2) and illustrate the value of the proposed improvements in MAGNets.

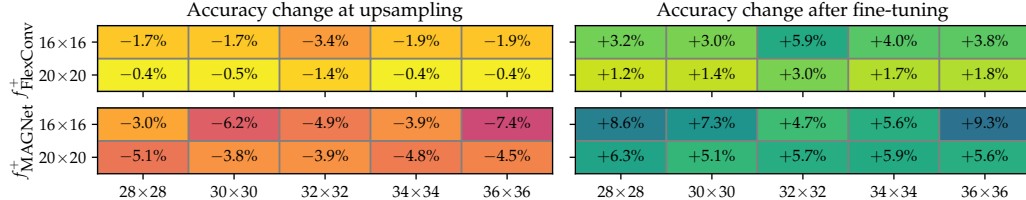

Figure 5: Alias-free FlexNet-16 on CIFAR-10. We report change in accuracy between source and target resolutions, directly after upsampling (left) and after fine-tuning (right) (means over five runs).

## 4.3 ALIAS-FREE FLEXNETS

**Regularizing the FlexConv mask.** Though including $f_{w_{\text{gauss}}}^+$ in the frequency analysis of MAGNets is crucial for the accuracy of the derivation, including the FlexConv mask in aliasing regularization is undesirable, as it steers the model to learn large kernels in order to minimize the loss (see Eq. 25). However, excluding the mask from regularization could compromise the ability of FlexNet to generalize to higher resolutions. Here, we experiment with this trade-off.

Figure 5 shows accuracy change between ten source and target resolution combinations on CIFAR-10, both for including and excluding the FlexConv mask in the aliasing regularization. We train at the source resolution for 100 epochs, before testing the model at the target resolution with the upsampling described in Sec. 3.3. Next, we adjust $f_{\text{Nyq}}(k)$ to the target resolution, and finetune each model for 100 epochs at the target resolution.

We find that regularizing just $f_{\text{MAGNet}}^+$ yields a trade-off. It increases the accuracy difference between low and high resolution inference, but also increases the fine-tune accuracy at the target resolution. We therefore choose to, by default, regularize $f_{\text{MAGNet}}^+$ only.

Table 4: Alias-free FlexNets on CIFAR-10.

| MODEL | SIZE | CIFAR-10 ACC. | |
|---|---|---|---|
| | | 16 px | $\Delta_{16\text{px}}$ 32 px |
| CIFARResNet-44 | 0.66M | 85.8 ± 0.2 | -31.6 ± 1.3 |
| FlexNet-16 w/ conv. ($k = 3$) | 0.17M | 85.3 ± 0.2 | -21.2 ± 1.0 |
| FlexNet-16 w/ conv. ($k = 33$) | 20.0M | 67.7 ± 0.6 | -57.1 ± 1.6 |
| FlexNet-16 w/ N-Jets | 0.70M | **86.4** ± 0.2 | -5.5 ± 1.3 |
| CKCNN-16$_{\text{SIREN}}$ | 0.63M | 45.9 ± 1.0 | -15.8 ± 1.2 |
| FlexNet-16$_{\text{SIREN}}$ | 0.63M | 70.4 ± 0.8 | -50.0 ± 16.9 |
| FlexNet-16 w/o reg. | 0.67M | **86.4** ± 0.4 | -34.4 ± 14.3 |
| FlexNet-16 w/ reg. $f_{\text{MAGNet}}^+$ | 0.67M | **86.5** ± 0.1 | -3.8 ± 2.0 |
| FlexNet-16 w/ reg. $f_{\text{FlexConv}}^+$ | 0.67M | 85.1 ± 0.3 | **-3.3** ± 0.3 |

Results of our alias-free FlexNet training on CIFAR-10 are in Table 4. We observe that the performance of a FlexNet trained without aliasing regularization largely breaks down when the dataset is upscaled. However, with our aliasing regularization most of the performance is retained.

Comparatively, FlexNet retains more of the source resolution performance than FlexNets with N-Jet layers, while baselines degrade drastically at the target resolution. Fig. 8 shows the effect of aliasing regularization on the frequency components of FlexConv.

**Training at lower resolutions saves compute.** We can train alias-free FlexNets at lower resolutions. To verify that this saves compute, we time the first 32 batches of training a FlexNet-7 on CIFAR-10. We compare against training on $16 \times 16$ images (downsampled before training). On 16x16 images, each batch takes 179ms (± 7ms). On 32x32 images, each batch takes 222ms (± 9ms). Therefore, we save 24% training time when training FlexNets alias-free at half the native CIFAR-10 resolution.

## 5 DISCUSSION

**Learned kernel sizes match conventional priors.** Commonly, CNNs use architectures of small kernels and pooling layers. This allows convolutions to build a progressively growing receptive field. With learnable kernel sizes, FlexNet could learn a different prior over receptive fields, e.g., large kernels first, and small kernels next. However, FlexNets learn to increase kernel sizes progressively (Fig. 6), and match the network design that has been popular since AlexNet (Krizhevsky et al., 2012).

**Mask initialization as a prior for feature importance.** The initial values of the FlexConv mask can be used to prioritize information at particular input regions. For instance, initializing the center of mask on the first element of sequential FlexConvs can be used to prioritize information from the far past. This prior is advantageous for tasks such as npCIFAR10. We observe that using this prior on npCIFAR10 leads to much faster convergence and better results (68.33% acc. w/ FlexTCN-2).

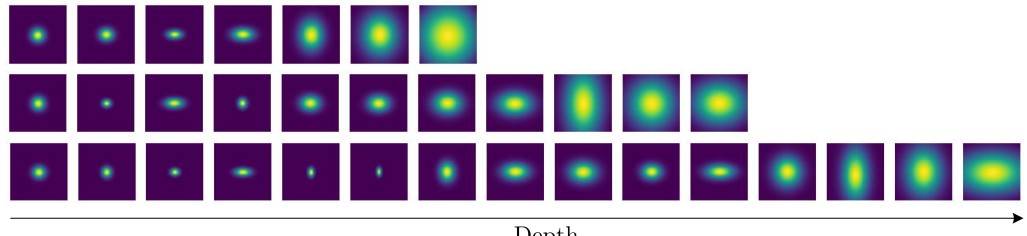

Figure 6: Learned FlexConv masks for FlexNets with 3, 5 and 7 residual blocks. FlexNets learn very small kernels at shallow layers, which become larger as a function of depth.

**MAGNet regularization as prior induction.** MAGNets allow for analytic control of the properties of the resulting representations. We use this property to generate alias-free kernels. However, other desiderata could be induced, e.g., smoothness, for the construction of implicit neural representations.

**Benefits of cropping and the influence of `PyTorch`.** Dynamic cropping adjust the computational cost of the convolutions on the fly. For a signal of size $M$ and a cropped kernel size $k$, this incurs in savings from $O(M^{2^D})$ to $O(M^D k^D)$ relative to using global kernel sizes ($O(M^4)$ to $O(M^2 k^2)$ in 2D). We test this theoretical speed up in a controlled environment for the Speech Commands and CIFAR-10 datasets. Cropping reduces the per-epoch run time by a factor of 11.8x and 5.5x for Speech Commands and CIFAR-10, respectively. Interestingly, however, both run times become similar if the flag `torch.backends.cudnn.benchmark` is activated, with global kernel sizes being sometimes faster. This is because this flag tells `PyTorch` to optimize the convolution algorithms used under the hood, and some of these CUDA algorithms seem to be faster than our masking strategy on `Python`.

## 6 LIMITATIONS

**Dynamic kernel sizes: computation and memory cost of convolutions with large kernels.** Performing convolutions with large convolutional kernels is a compute-intensive operation. FlexConvs are initialized with small kernel sizes and their inference cost is relatively small at the start of training. However, despite the cropping operations used to improve computational efficiency (Figs. 1, 3, Tab. 3), the inference time may increase to up to double as the learned masks increase in size. At the cost of more memory, convolutions can be sped up by performing them in the frequency domain. However, we observe that this does not bring gains for the image data considered because FFT convolutions are faster only for very large convolutional kernels (in the order of hundreds of pixels).

**Remaining accuracy drop in alias-free FlexNets.** Some drop in accuracy is still observed when using alias-free FlexNets at a higher test resolutions (Tab. 4). Although more evidence is needed, this may be caused by aliasing effects introduced by ReLU (Vasconcelos et al., 2021), or changes in the activation statistics of the feature maps passed to global average pooling (Touvron et al., 2019).

## 7 CONCLUSION

We propose FlexConv, a convolutional operation able to learn high bandwidth convolutional kernels of varying size during training at a fixed parameter cost. We demonstrate that FlexConvs are able to model long-term dependencies without the need of pooling, and shallow pooling-free FlexNets achieve state-of-the-art performance on several sequential datasets, match performance of recent works with learned kernel sizes with less compute, and are competitive with much deeper ResNets on image benchmark datasets. In addition, we show that our alias-free convolutional kernels allow FlexNets to be deployed at higher resolutions than seen during training with minimal precision loss.

**Future work.** MAGNets give control over the bandwidth of the kernel. We anticipate that this control has more uses, such as fighting sub-sampling aliasing (Zhang, 2019; Kayhan & Gemert, 2020; Karras et al., 2021). With the ability to upscale FlexNets to different input image sizes comes the possibility of transfer learning representations between previously incompatible datasets, such as CIFAR-10 and Imagenet. In a similar vein, the automatic adaptation of FlexConv to the kernel sizes required for the task at hand may make it possible to generalize the FlexNet architecture across different tasks and datasets. Neural architecture search (Zoph & Le, 2016) could see benefits from narrowing the search space to exclude kernel size and pooling layers. In addition, we envisage additional improvements from structural developments of FlexConvs such as attentive FlexNets.

## REPRODUCIBILITY STATEMENT

We hope to inspire others to use and reproduce our work. We publish the source code of this work, for which the link is provided in Sec. 4.2. Sec. 4 and Appx. D.1 detail FlexNet, its hyperparameters and optimization procedure. The full derivation of the aliasing regularization objective is included in Appx. A.1. We report means over multiple runs for many experiments, to ensure the reported results are fair and reproducible, and do not rely on tuning of the random seed. All datasets used in our experiments are publicly available. If any questions remain, we welcome one and all to contact the corresponding author.

## ACKNOWLEDGMENTS

We thank Nergis Tömen for her valuable insights regarding signal processing principles for FlexConv, and Silvia-Laura Pintea for explanations and access to code of her work Pintea et al. (2021). We thank Yerlan Idelbayev for the use of the CIFARResNet code.

This work is co-supported by the Qualcomm Innovation Fellowship granted to David W. Romero. David W. Romero sincerely thanks Qualcomm for his support. David W. Romero is financed as part of the Efficient Deep Learning (EDL) programme (grant number P16-25), partly funded by the Dutch Research Council (NWO). Robert-Jan Bruintjes is financed by the Dutch Research Council (NWO) (project VI.Vidi.192.100). All authors sincerely thank everyone involved in funding this work.

This work was partially carried out on the Dutch national infrastructure with the support of SURF Cooperative. We used Weights & Biases (Biewald, 2020) for experiment tracking and visualizations.

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

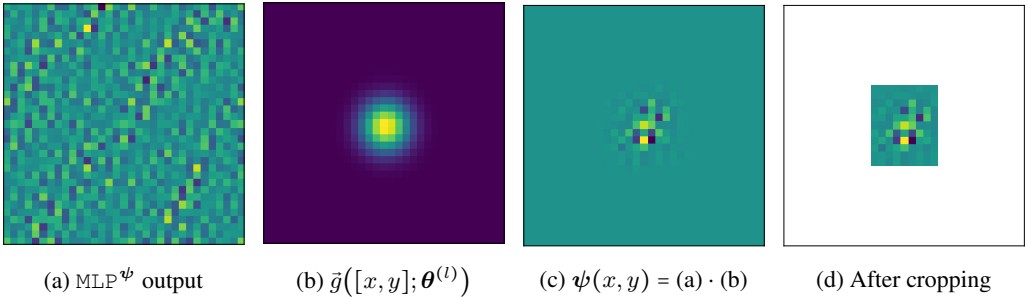

(a) $\mathtt{MLP}^{\psi}$ output     (b) $\vec{g}\big([x,y];\boldsymbol{\theta}^{(l)}\big)$     (c) $\psi(x,y) = $ (a) $\cdot$ (b)     (d) After cropping

Figure 7: Example kernels, generated step by step. FlexConv samples a kernel from $\mathtt{MLP}^{\psi}$ (a), which is attenuated by an anistropic Gaussian envelope with learned parameters $\boldsymbol{\theta}^{(l)}$ (b), creating (c) which is cropped to contain only values of $> 0.1$ (d).

## A   ALIAS-FREE FLEXCONV REGULARIZATION

In this section we provide the complete derivation and analysis for our FlexConv regularization against aliasing. First, we derive the analytic maximum frequency component of a FlexConv kernel. Next, we compute the Nyquist frequency of a FlexConv kernel, and subsequently show how to combine the previous results into a regularization term to train alias-free FlexConvs.

### A.1   ANALYZING THE FREQUENCY SPECTRUM OF FLEXCONV

In order to make FlexConv alias-free (Sec. 3.3), we need to compute the maximum frequency component of the kernels generated by a MAGNet, so that we can regularize it during training. In this section we analytically derive this maximum frequency component from the parameters of the MAGNet.

Recall that MAGNets generate a kernel $\psi(x,y)$ through of a succession of anisotropic Gabor filters and linear layers (Sec. 3.2, Eqs. 2–7):

$$\mathbf{h}^{(1)} = \mathbf{g}\big([x,y];\boldsymbol{\theta}^{(1)}\big) \qquad\qquad \mathbf{g} : \mathbb{R}^2 \to \mathbb{R}^{N_{\mathrm{hid}}}$$

$$\mathbf{h}^{(l)} = \big(\mathbf{W}^{(l)}\mathbf{h}^{(l-1)} + \mathbf{b}^{(l)}\big) \cdot \mathbf{g}\big([x,y];\boldsymbol{\theta}^{(l)}\big) \qquad \mathbf{W}^{(l)} \in \mathbb{R}^{N_{\mathrm{hid}} \times N_{\mathrm{hid}}}, \mathbf{b}^{(l)} \in \mathbb{R}^{N_{\mathrm{hid}}}$$

$$\psi(x,y) = \mathbf{W}^{(L)}\mathbf{h}^{(L-1)} + \mathbf{b}^{(L)} \qquad \mathbf{W}^{(L)} \in \mathbb{R}^{(N_{\mathrm{out}} \times N_{\mathrm{in}}) \times N_{\mathrm{hid}}}, \mathbf{b}^{(L)} \in \mathbb{R}^{(N_{\mathrm{out}} \times N_{\mathrm{in}})}$$

$$\mathbf{g}\big([x,y];\boldsymbol{\theta}^{(l)}\big) = \exp\left( -\frac{1}{2}\Big[\big(\gamma_{\mathrm{X}}^{(l)}\big(x - \boldsymbol{\mu}_{\mathrm{X}}^{(l)}\big)\big)^2 + \big(\gamma_{\mathrm{Y}}^{(l)}\big(y - \boldsymbol{\mu}_{\mathrm{Y}}^{(l)}\big)\big)^2\Big] \right) \mathrm{Sin}\big(\mathbf{W}_{\mathrm{g}}^{(l)}[x,y] + \mathbf{b}_{\mathrm{g}}^{(l)}\big)$$

$$\boldsymbol{\theta}^{(l)} = \Big\{ \gamma_{\mathrm{X}}^{(l)} \in \mathbb{R}^{N_{\mathrm{hid}}}, \gamma_{\mathrm{Y}}^{(l)} \in \mathbb{R}^{N_{\mathrm{hid}}}, \boldsymbol{\mu}_{\mathrm{X}}^{(l)} \in \mathbb{R}^{N_{\mathrm{hid}}}, \boldsymbol{\mu}_{\mathrm{Y}}^{(l)} \in \mathbb{R}^{N_{\mathrm{hid}}}, \mathbf{W}_{\mathrm{g}}^{(l)} \in \mathbb{R}^{N_{\mathrm{hid}} \times 2}, \mathbf{b}_{\mathrm{g}}^{(l)} \in \mathbb{R}^{N_{\mathrm{hid}}} \Big\}$$

To analyse the maximum frequency component $f_{\mathrm{MAGNet}}^{+}$, we analyse the frequency components of the Gabor filters used in MAGNet, and retain their maximum. We then plug the found frequency component into the analysis of Fathony et al. (2021) to show how the frequency responses of Gabor filters and linear layers interact in MFNs. Finally, we add the effect of the FlexConv Gaussian mask to our analysis to obtain the maximum frequency component ot the final FlexConv kernel $f_{\mathrm{FlexConv}}^{+}$.

**Sine term in Gabor filters.** In a Gabor filter, the sine term is multiplied with a Gaussian envelope. The frequency (in radians) of a sine function of the form $\mathrm{Sin}(\boldsymbol{w}^T[x,y] + b)$ is given by $\boldsymbol{w}$. We divide by $2\pi$ to convert the frequency units to Hertz, for compatibility with the rest of the analysis. For 2D inputs, the maximum frequency component of the sine function correspond to the largest frequency in the two input dimensions:

$$f_{\mathrm{Sin}}^{+} = \max_{j} \frac{w_j}{2\pi}. \tag{10}$$

The sine terms in MAGNets have multiple output channels: $\mathrm{Sin}\big(\mathbf{W}_{\mathrm{g}} \cdot [x,y] + \mathbf{b}_{\mathrm{g}}^{(l)}\big)$. Effectively, we compute the sine term independently for each channel:

$$f_{\mathrm{Sin},i}^{+} = \max_{j} \frac{\mathbf{W}_{\mathrm{g},i,j}}{2\pi}. \tag{11}$$

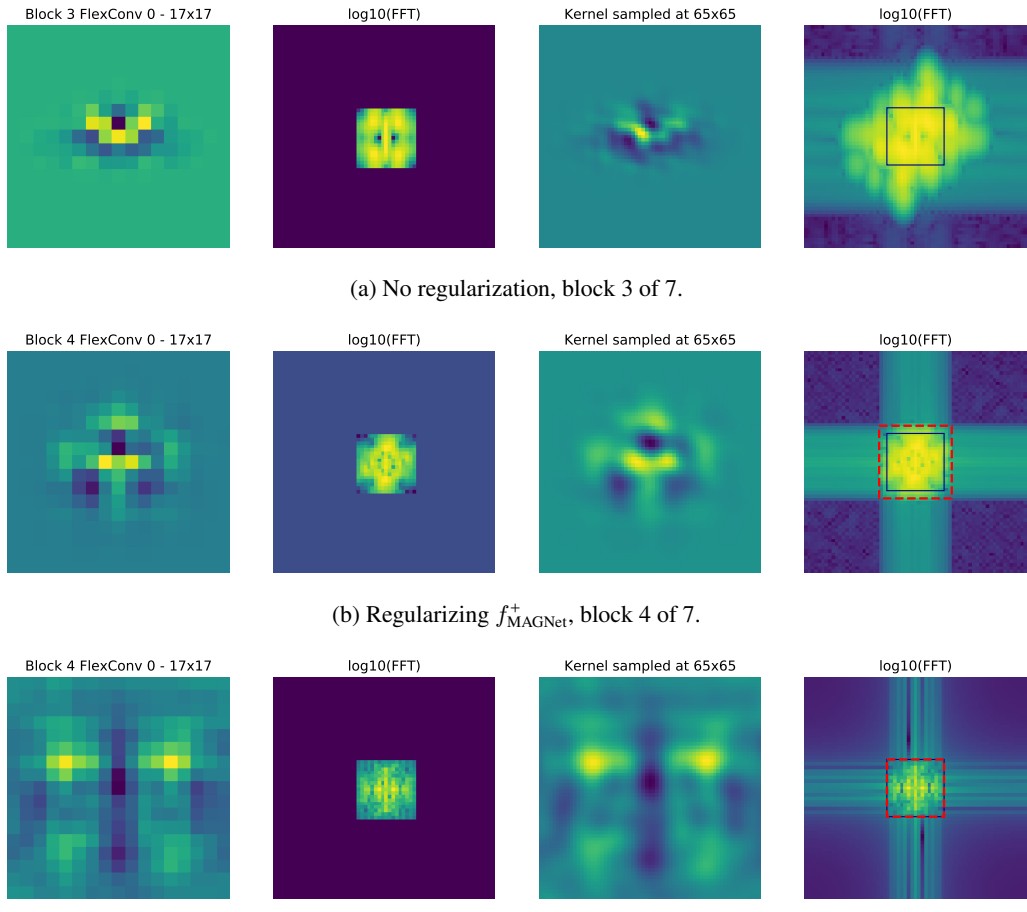

(a) No regularization, block 3 of 7.

(b) Regularizing $f_{\mathrm{MAGNet}}^+$, block 4 of 7.

(c) Regularizing $f_{\mathrm{FlexConv}}^+$, block 4 of 7.

Figure 8: Example kernels from FlexNet-16 models trained (i) without regularization, (ii) with aliasing regularization of $f_{\mathrm{MAGNet}}^+$, (iii) with aliasing regularization of $f_{\mathrm{FlexConv}}^+$. In the columns, from left to right: (i) original kernel at $33 \times 33$, (ii) FFT of the original kernel, (iii) kernel inferred at $65 \times 65$, to find aliasing effects, (iiii) FFT of the $65 \times 65$ kernel, with the solid line showing the Nyquist frequency of the $33 \times 33$ kernel, and the red dotted line showing the maximum frequency component as computed by our analysis. For $f_{\mathrm{FlexConv}}^+$ the maximum frequency matches almost exactly with the Nyquist frequency, showing that our aliasing regularization works. For $f_{\mathrm{MAGNet}}^+$, the maximum frequency is slightly higher than the Nyquist frequency, as the FlexConv mask is not included in the frequency term derivation. This is reflected in the slightly worse resolution generalization results reported in Sec. 4.3. Furthermore, some aliasing effects are still apparent for the aliasing regularized models, as discussed in Sec. 6.

**Gaussian term in Gabor filters.** In a Gabor filter, a Gaussian envelope modulates a sine term. Let us assume for now that the Gaussian envelope is isotropic, rather than anisotropic as in MAGNets, and has single-channel output. By applying the convolution theorem, the sine term is equivalently convolved with the Fourier transform of the Gaussian envelope in the frequency domain. Since the Fourier transform of a Gaussian envelope is another Gaussian envelope, the application of a Gaussian envelope amounts to blurring with a Gaussian kernel in the frequency domain. The size of the envelope in the Fourier domain $\sigma_F$ can be derived from the standard deviation of the Gaussian envelope in the spatial domain $\sigma_T$ as follows:

$$\sigma_T \sigma_F = \frac{1}{2\pi} \Rightarrow \sigma_F = \frac{1}{2\pi\sigma_T}. \tag{12}$$

Gaussian blurs induce impulse signals to have a long tail. Consequently, we must define a cutoff point for this tail in terms of standard deviations to derive the maximum added frequency induced by the blur. We describe the cutoff point as $\sigma_{\text{cut}} \in \mathbb{N}$. Typical choices for $\sigma_{\text{cut}}$ are known as the *empirical*, or the "68-95-99.7" rule (Hald, 2007). We choose a standard of two standard deviations, i.e., $\sigma_{\text{cut}}=2$, which covers 95% of the mass of the Gaussian envelope.

For an isotropic Gabor filter with $\gamma=\sigma_T^{-1}$, the maximum frequency of its Gaussian envelope $f_{\text{env}}^+$ is:

$$f_{\text{env}}^+ = \frac{\sigma_{\text{cut}}}{2\pi(\sigma_T)^{-1}} = \frac{\sigma_{\text{cut}}\gamma}{2\pi}. \tag{13}$$

**Anisotropic envelopes.** Our analysis so far assumes an isotropic Gaussian envelope in the Gabor filter. However, we need to account for the anisotropic Gaussian envelopes in MAGNets. Anisotropic filters have not one but two $\gamma$ parameters: $\{\gamma_X, \gamma_Y\}$. The smallest of these will contribute most to $f_{\text{env}}^+$, as it will blur the most, so it is sufficient to compute $f_{\text{env}}^+$ only using the smallest of the two $\gamma$ terms:

$$f_{\text{env}}^+(\gamma_X, \gamma_Y) = f_{\text{env}}^+(\min\{\gamma_X, \gamma_Y\}). \tag{14}$$

The other assumption we made before was to work with single-channel outputs. MAGNets however use multi-channel outputs with independent Gaussian terms. The maximum frequency of multi-channel Gaussian envelopes is given by:

$$f_{\text{env},i}^+(\boldsymbol{\gamma}_X, \boldsymbol{\gamma}_Y) = f_{\text{env}}^+\left(\min\{\gamma_{X,i}, \gamma_{Y,i}\}\right) = \frac{\sigma_{\text{cut}}\min\{\gamma_{X,i}, \gamma_{Y,i}\}}{2\pi}, \tag{15}$$

where the subscript $i$ indexes the channels of the multi-channel Gaussian envelopes.

**Maximum frequency component of anisotropic Gabor filters.** Finally, the maximum frequency component of the $i$-th channel of an anisotropic Gabor filter $\mathbf{g}$ is given by:

$$\begin{aligned} f_{\text{Gabor},i}^+ &= f_{\text{Sin},i}^+(\mathbf{W}_{\text{g}}) + f_{\text{env},i}^+(\boldsymbol{\gamma}_X, \boldsymbol{\gamma}_Y) \\ &= \left(\max_j \frac{\mathbf{W}_{\text{g},i,j}}{2\pi}\right) + \frac{\sigma_{\text{cut}}\min\{\gamma_{X,i}, \gamma_{Y,i}\}}{2\pi}. \end{aligned} \tag{16}$$

Figure 9 illustrates the frequency spectrum of an example Gabor filter.

**Maximum frequency component of a MAGNet.** Fathony et al. (2021) characterize the expansion of each term of the isotropic Gabor layers in MFNs in the final MFN output. In Eq. 25, Fathony et al. (2021) demonstrate that the MFN representation contains a set of sine frequencies $\overline{\boldsymbol{\omega}}$ given by:

$$\overline{\boldsymbol{\omega}} = \left\{ s_L \omega_{i_L}^{(L)} + s_{L-1} \omega_{i_{L-1}}^{(L-1)} + \cdots + s_l \omega_{i_2}^{(2)} + \omega_{i_1}^{(1)} \right\}. \tag{17}$$

Here, the indexes $i_1, i_2, \cdots, i_{L-1}$ range over all possible indices of each hidden unit of each layer of an MFN, and $s_2, \cdots, s_L \in \{-1, +1\}$ range over all $2^{L-1}$ possible binary signs. In other words, Fathony et al. (2021) demonstrate that the representation of an MFN at a particular layer contains an exponential combination of all possible positive and negative combinations of the frequencies of the sine terms in each hidden unit at each layer in the MFN up to the current layer.

The original analysis uses these terms to argue that MFNs model exponentially many terms through a linear amount of layers. For our purpose of computing the frequency response of the MAGNet generated kernel, we can plug our derivation of the frequencies of the Gabor filter $f_{\text{Gabor}}$ into $\overline{\boldsymbol{\omega}}$ to compute the frequency spectrum of the generated kernel:

$$\boldsymbol{f}_{\text{MAGNet}}^+ = \left\{ s_L f_{\text{Gabor},i_L}^{(L)} + s_{L-1} f_{\text{Gabor},i_{L-1}}^{(L-1)} + \cdots + s_2 f_{\text{Gabor},i_2}^{(2)} + f_{\text{Gabor},i_1}^{(1)} \right\} \tag{18}$$

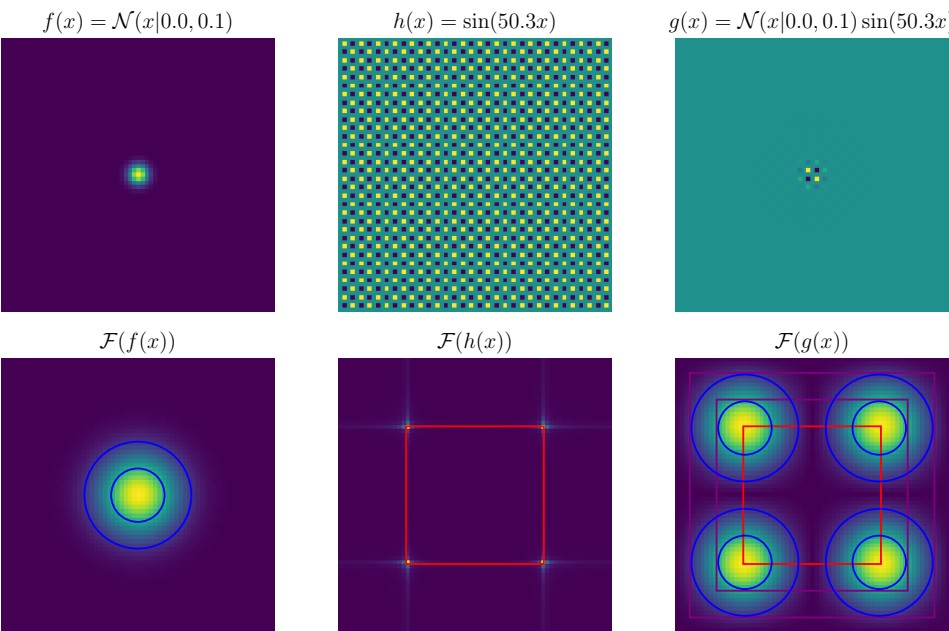

Figure 9: Decomposition of a Gabor filter and its frequency spectrum. Top row: a decomposition of a Gabor filter (right) into its Gaussian term (left) and its sine term (center). Bottom row: frequency responses for each respective filter. The Fourier transform of a Gaussian envelope is a Gaussian envelope (blue circles show $\sigma_{\mathcal{F}}$ for $h = \{1, 2\}$). The Fourier transform of a sine pattern is a collection of symmetrical impulse signals (red box shows the Nyquist frequency). The Gaussian envelope blurs the frequency response of the sine term (purple boxes show the frequency response for $h = \{1, 2, 3\}$).

As stated before, we are only interested in the maximum frequency in the frequency spectrum. We can therefore simplify Eq. 18 in two ways. First, we simplify over MAGNet layers by taking the maximum value of the spectrum, which is the sum over all layers using only the positive binary signs in $s_L$ (Eq. 19). Next, we simplify over channel indices by retaining only the channel index that results in the highest frequency (Eq. 20). The maximum frequency of a MAGNet is shown in Eq. 21:

$$\boldsymbol{f}^+_{\text{MAGNet}} = \left\{ (+1) f^{+\,(L)}_{\text{Gabor},i_L} + (+1) f^{+\,(L-1)}_{\text{Gabor},i_{L-1}} + \cdots + (+1) f^{+\,(2)}_{\text{Gabor},i_2} + f^{+\,(1)}_{\text{Gabor},i_1} \right\} \tag{19}$$

$$= \left\{ f^{+\,(L)}_{\text{Gabor},i_L} + f^{+\,(L-1)}_{\text{Gabor},i_{L-1}} + \cdots + f^{+\,(2)}_{\text{Gabor},i_2} + f^{+\,(1)}_{\text{Gabor},i_1} \right\}$$

$$f^+_{\text{MAGNet}} = \max_{i_L} \left( f^{+\,(L)}_{\text{Gabor},i_L} \right) + \max_{i_{L-1}} \left( f^{+\,(L-1)}_{\text{Gabor},i_{L-1}} \right) \cdots + \max_{i_2} \left( f^{+\,(2)}_{\text{Gabor},i_2} \right) + \max_{i_1} \left( f^{+\,(1)}_{\text{Gabor},i_1} \right) \tag{20}$$

$$= \sum_{l=1}^{L} \max_{i_l} \left( f^{+\,(l)}_{\text{Gabor},i_l} \right)$$

$$= \sum_{l=1}^{L} \max_{i_l} \left( \left( \max_{j} \frac{\mathbf{W}^{(l)}_{\text{g},i_l,j}}{2\pi} \right) + \frac{\sigma_{\text{cut}} \min\{\boldsymbol{\gamma}^{(l)}_{\text{X},i_l}, \boldsymbol{\gamma}^{(l)}_{\text{Y},i_l}\}}{2\pi} \right). \tag{21}$$

**Effect of the Gaussian mask in the frequency components of a FlexConv.** FlexConvs attenuate the MAGNet output with a Gaussian mask. The Gaussian mask (Eq. 1) works analogously to the Gaussian envelope term in the Gabor filter: it blurs the frequency components of the generated kernel with standard deviation $\sigma_{\text{F}}$. Therefore, we can reuse our derivation for the Gaussian envelope of the

Gabor filter (Eq. 15). The maximum frequency component of a FlexConv kernel is given by:

$$f^+_{\text{FlexConv}} = f^+_{\text{MAGNet}} + f^+_{\text{env}}$$

$$= f^+_{\text{MAGNet}} + \frac{\sigma_{\text{cut}} \min\{\sigma_X^{-1}, \sigma_Y^{-1}\}}{2\pi} = f^+_{\text{MAGNet}} + \frac{\sigma_{\text{cut}}}{\max\{\sigma_X, \sigma_Y\}2\pi}$$

$$= \sum_{l=1}^{L} \max_{i_l} \left( \left( \max_j \frac{\mathbf{W}^{(l)}_{g,i_l,j}}{2\pi} \right) + \frac{\sigma_{\text{cut}} \min\{\gamma^{(l)}_{X,i_l}, \gamma^{(l)}_{Y,i_l}\}}{2\pi} \right) + \frac{\sigma_{\text{cut}}}{\max\{\sigma_X, \sigma_Y\}2\pi}. \quad (22)$$

**Visualization of regularized kernels.** Fig. 8 shows example kernels from FlexNets trained with aliasing regularization. The frequency domain plots confirm the accuracy of our frequency component regularization.

## A.2  REGULARIZING THE FREQUENCY RESPONSE OF FLEXCONV

**Nyquist frequency of a FlexConv kernel.** Given the sampling rate $f_s$ of the kernel, we can compute its Nyquist frequency $f_{\text{Nyq}}$ as:

$$f_{\text{Nyq}} = \frac{1}{2} f_s \quad (23)$$

To compute the sampling rate, we note that the kernel coordinates input to our MAGNet stretch over a $[-1, 1]^D$ domain. For a kernel of length $k$, we therefore sample one point in every $f_s = \frac{k-1}{2}$ units.

Knowing the sampling rate in terms of the kernel size allows us to express the Nyquist frequency in terms of the (pre-masked) kernel size:

$$f_{\text{Nyq}}(k) = \frac{1}{2} \frac{k-1}{2} = \frac{k-1}{4}. \quad (24)$$

Note that the kernel size in a FlexConv is initialized to be equal to the resolution of the data, if it is odd. For even resolutions, it corresponds to the resolution of the data plus one.

**Constructing the regularization term.** We train FlexConv with a regularization term on the frequency response of the generated kernel to ensure that aliasing effects do not distort the performance of the model when it is inferred at a higher resolution. This section details the implementation of the regularization function.

From the parameters of each FlexConv module, we compute $f^+_{\text{FlexConv}}$ according to Eq. 22. For the amount of standard deviations to use in determining $f^+_{\text{env}}$ (Eq. 15) we use $h = 2$. From the kernel size $k$ of the FlexConv module we compute $f_{\text{Nyq}}(k)$ according to Eq. 24. We then apply an L2 regularizer over the amount that $f^+_{\text{FlexConv}}$ exceeds $f_{\text{Nyq}}(k)$:

$$\mathcal{L}_{\text{HF}} = \| \max\{f^+_{\text{FlexConv}}, f_{\text{Nyq}}(k)\} - f_{\text{Nyq}}(k) \|^2. \quad (25)$$

We weight $\mathcal{L}_{\text{HF}}$ by $\lambda = 0.1$ when adding it to our loss function.

**Improved implementation.** Eq. 25 contains a sum over the L layers of the MAGNet. In practice, we prefer to regularize each layer $l \in L$ separately, so that the gradients of the regularization of different layers are not dependent on each other. We therefore implement the anti-aliasing regularization by regularizing each MAGNet layer independently, and spreading the $f^+_{\text{env}}$ term from the gaussian mask uniformly over all MAGNet layers:

$$\mathcal{L}_{\text{HF},l} = \| \max \left\{ f^+_{\text{MAGNet},l} + \frac{f^+_{\text{env}}}{L}, \frac{f_{\text{Nyq}}(k)}{L} \right\} - \frac{f_{\text{Nyq}}(k)}{L} \|^2 \quad (26)$$

$$= \| \max \left\{ \max_{i_l} \left( f^{+\,(l)}_{\text{Gabor},i_l} \right) + \frac{f^+_{\text{env}}}{L}, \frac{f_{\text{Nyq}}(k)}{L} \right\} - \frac{f_{\text{Nyq}}(k)}{L} \|^2. \quad (27)$$

In the code, we refer to this method as the `together` method, versus the `summed` method of Eq. 25. In preliminary experiments, we observed improved performance of anti-aliasing training when using the `together` method. All of our experiments anti-aliasing experiments therefore use the `together` setting.

# B    DATASET DESCRIPTION

## B.1    IMAGE FITTING DATASETS

**Kodak dataset.** The Kodak dataset (Kodak, 1991) consists of 24 natural images of size $768 \times 512$. This dataset is a popular benchmark used for compression and image fitting methods.

## B.2    SEQUENTIAL DATASETS

**Sequential and Permuted MNIST.** The sequential MNIST dataset (sMNIST) (Le et al., 2015)takes the 28×28 images from the original MNIST dataset (LeCun et al., 1998), and presents them as a sequence of 784 pixels. The goal of this task is to perform digit classification given the representation of the last sequence element of a sequential model. Consequently, good predictions require the model to preserve long-term dependencies up to 784 steps in the past.

The permuted MNIST dataset (pMNIST) additionally changes the order of all the sMNIST sequences by a random permutation. Consequently, models can no longer rely on local features to construct good feature representations. As a result, the classification problem becomes more difficult, and the importance of long-term dependencies more pronounced.

**Sequential and Noise-Padded CIFAR10.** The sequential CIFAR10 dataset (sCIFAR10) (Chang et al., 2017) takes the 32×32 images from the original CIFAR10 dataset (Krizhevsky et al., 2009) and presents them as a sequence of 1,024 pixels. The goal of this task is to perform image classification given the representation of the last sequence element of a sequential model. This task is more difficult than sMNIST, as a larger memory horizon is required to solve the task and more complex structures and intra-class variations are present in the data (Bai et al., 2018b).

The noise-padded CIFAR10 dataset (npCIFAR10) (Chang et al., 2019) flattens the images from the original CIFAR10 dataset (Krizhevsky et al., 2009) along their rows to create a sequence of length 32, and 96 channels (32 rows × 3 channels). Next, these sequences are concatenated with 968 entries of noise to form the final sequences of length 1000. As for sCIFAR10, the goal of the task is to perform image classification given the representation of the last sequence element of a sequential model.

**CharacterTrajectories.** The CharacterTrajectories dataset is part of the UEA time series classification archive (Bagnall et al., 2018). It consists of 2858 time series of length 182 and 3 channels representing the $x, y$ positions, and the tip force of a pen while writing Latin alphabet characters in a single stroke. The goal is to classify out of 20 classes the written character using the time series data.

**Speech Commands.** The Speech Commands dataset (Warden, 2018) consists of 105,809 one-second audio recordings of 35 spoken words sampled at 16kHz. Following Kidger et al. (2020), we extract 34975 recordings from ten spoken words to construct a balanced classification problem. We refer to this dataset as **SpeechCommands_raw**, or **SC_raw** for short. Furhtermore, we utilize the preprocessing steps of Kidger et al. (2020) and extract mel-frequency cepstrum coefficients from the raw data. The resulting dataset, abreviated **SC**, consists of time series of length 101, and 20 channels.

## B.3    IMAGE BENCHMARK DATASETS

**MNIST.** The MNIST hadwritten digits datset (LeCun & Cortes, 2010) consists of 70,000 gray-scale handwritten digits of size 28×28, divided into a training and test sets of 60,000 and 10,000 images, respectively. The goal of the task is to classify these digits as one of the ten possible digits $(0, 1, ..8, 9)$.

**CIFAR-10** The CIFAR-10 dataset (Krizhevsky et al., 2009) consists of 60,000 natural images from 10 classes of size 32×32, divided into training and test sets of 50,000 and 10,000 images, respectively.

**STL-10.** The STL-10 dataset (Coates et al., 2011) is a subset of the ImageNet dataset (Krizhevsky et al., 2012) consisting of 5,000 natural images from 10 classes of size 96×96, divided into trainint and test sets of 4,500 and 500 images, respectively.

**ImageNet-**k**.** The Imagenet-k (Chrabaszcz et al., 2017) dataset is derived from the ImageNet dataset Russakovsky et al. (2015) by downsampling all samples to a resolution k $\in [64, 32, 16, 8]$. The dataset contains 1000 classes with 1,281,167 training samples and 50,000 validation samples.

Table 5: Average PSNR for fitting of images in the Kodak dataset. Both our improved initialization scheme, as well as the inclusion of anisotropic Gabor functions lead to better reconstructions.

| MODEL | # PARAMS | IMPROVED INIT | PSNR |
|---|---|---|---|
| SIREN | 7.14K | - | 25.665 |
| MFN$_{Fourier}$ | 7.40K | - | 23.276 |
| MFN$_{Gabor}$ | 7.11K | ✗ | 25.361 |
| | | ✓ | **25.606** |
| MAGNet | 7.36K | ✗ | 25.791 |
| | | ✓ | **25.893** |

Table 6: Full results on CIFAR-10. We report results over three runs per setting. CIFARResNet-44 w/ CKConv is a CIFARResNet-44 where all convolutional layers are replaced with CKConvs with $k = 3$. CIFARResNet-44 w/ FlexConv is a CIFARResNet-44 where all convolutional layers are replaced with FlexConv with learned kernel size, except for the shortcut connections of the strided convolutional layers, which are pointwise convolutions. *Results are taken from the respective original works instead of reproduced. †Results are from single run.

| MODEL | SIZE | CIFAR-10 ACC. |
|---|---|---|
| DCN-$\sigma^{ji}$ (Tomen et al., 2021) | 0.47M | 89.7 ± 0.3* |
| N-Jet-CIFARResNet32 (Pintea et al., 2021) | 0.52M | 92.3 ± 0.3* |
| N-Jet-ALLCNN (Pintea et al., 2021) | 1.07M | 92.5 ± 0.1* |
| CIFARResNet-44 (He et al., 2016) | 0.66M | 92.9*† |
| CIFARResNet-44 (He et al., 2016) (our reproduction) | 0.66M | 90.9 ± 0.2 |
| CIFARResNet-44 w/ CKConv ($k = 3$) | 2.58M | 86.1 ± 0.9 |
| CIFARResNet-44 w/ FlexConv | 2.58M | 81.6 ± 0.8 |
| FlexNet-7 w/ conv. ($k = 3$) | 0.17M | 89.5 ± 0.3 |
| FlexNet-7 w/ conv. ($k = 33$) | 20.0M | 78.0 ± 0.3 |
| FlexNet-7 w/ N-Jet (Pintea et al., 2021) | 0.70M | 91.7 ± 0.1 |
| CKCNN$_{SIREN}$-3 | 0.26M | 72.4* |
| CKCNN$_{Fourier}$-3 | 0.27M | 83.8* |
| CKCNN$_{Gabor}$-3 | 0.28M | 85.6* |
| CKCNN$_{MAGNet}$-3 | 0.28M | 86.2* |
| CKCNN-7 | 0.63M | 71.7* |
| CKCNN$_{Fourier}$-7 | 0.63M | 84.6* |
| CKCNN$_{Gabor}$-7 | 0.67M | 87.7* |
| CKCNN$_{MAGNet}$-7 | 0.67M | 85.9* |
| FlexNet$_{SIREN}$-7 | 0.63M | 88.9* |
| FlexNet$_{Fourier}$-7 | 0.66M | 91.6* |
| FlexNet$_{Gabor}$-7 | 0.67M | 92.0* |
| FlexNet-3 | 0.27M | 90.4 ± 0.2 |
| FlexNet-5 | 0.44M | 91.0 ± 0.5 |
| FlexNet-7 | 0.67M | 92.2 ± 0.1 |

# C  ADDITIONAL EXPERIMENTS

## C.1  IMAGE CLASSIFICATION

**CIFAR-10.** Tab. 6 shows all results for our CIFAR-10 experiments, including more ablations.

**ImageNet-32.** Results for the ImageNet-32 experiment are shown in Table 7. FlexNets are slightly worse than CIFARResNet-32 (He et al., 2016) with slightly less parameters. However, the results

Table 7: Results on ImageNet-32. *Results are taken from the respective original works instead of reproduced. †Results are from a single run.

| MODEL | SIZE | IMAGENET-32 | |
|---|---|---|---|
| | | TOP-1 | TOP-5 |
| CIFARResNet-32 | 0.53M | 26.41 ± 0.13 | 49.37 ± 0.15 |
| WRN-28-1 | 0.44M | 32.03*† | 57.51*† |
| FlexNet-5 | 0.44M | 24.9 ± 0.4 | 47.7 ± 0.6 |

Table 8: Results for alias-free FlexNets on CIFAR-10 and ImageNet-k. $\Delta$ denotes difference in accuracy.

| MODEL | SIZE | IMAGENET-K TOP-1 | | |
| --- | --- | --- | --- | --- |
| | | $k = 16$ | $\Delta_{k=16}$ | $k = 32$ |
| CIFARResNet-32 | 0.52M | $16.1 \pm 0.0$ | $-11.6 \pm 0.4$ | |
| FlexNet-5 w/ N-Jets | 0.46M | $15.7 \pm 0.1$ | $-1.9 \pm 0.4$ | |
| FlexNet-5 | 0.44M | $14.9 \pm 0.1$ | $-1.9 \pm 1.7$ | |

Table 9: Results on MNIST. We train each model with three different seeds and report mean and standard deviation. *Results are taken from the respective original works instead of reproduced. †Results are from single run.

| MODEL | SIZE | MNIST ACC. |
| --- | --- | --- |
| Efficient-CapsNet (Mazzia et al., 2021) | 0.16M | **99.8**\*† |
| Network in Network (Lin et al., 2013) | N/A | 99.6\*† |
| VGG-5 (results from Kabir et al. (2020)) | 3.65M | 99.7\*† |
| FlexNet-16 | 0.67M | $99.7 \pm 0.0$ |

reported by Chrabaszcz et al. (2017) for Wide ResNets (Zagoruyko & Komodakis, 2016) outperform FlexNets by a significant margin.

**Alias-free ImageNet-32.** We report results for alias-free FlexNets on ImageNet-k (Chrabaszcz et al., 2017) in Table 8, to verify the results of alias-free training at a larger scale. We find that FlexConv and N-Jet both mostly retain classification accuracy between source and target resolution, while CIFARResNet-32 degrades drastically.

**MNIST and STL-10.** We additionally report results on MNIST (Tab. 9) and STL-10 (Tab. 10. We choose these dataset for the difference in image sizes of the training data. On MNIST, though performance on MNIST is quite saturated, we are competitive with state of the art methods. On STL-10 we are significantly worse than the baseline CIFARResNet from (Luo et al., 2020), though with significantly less parameters. We were not able to prepare a more relevant baseline for this experiment.

## D   EXPERIMENTAL DETAILS

### D.1   FLEXNET

We propose an image classification architecture named *FlexNet* (Fig. 10), consisting of a stack of FlexConv blocks followed by a global average pooling layer and a linear layer. FlexNets are named "FlexNet-L" where L indicates the amount of layers in the architecture.

**FlexBlock.** Each FlexBlock consists of two FlexConvs with BatchNorm (Ioffe & Szegedy, 2015) and dropout (Srivastava et al., 2014) ($d = 0.2$) as well as a residual connection. The width of a block $i$ is determined by scaling a base amount $c$ by progressively increasing factors: $c_i = [c, c \times 1.5, c \times 1.5, c \times 2.0, c \times 2.0](i)$. The default configuration of FlexNet uses $c = 22$. In FlexNet-N-Jet models, we scale $c$ to match the amount of parameters of the FlexNet in the comparison.

Table 10: Results on STL-10. We train each model with three different seeds and report mean and standard deviation. *Results are taken from Luo et al. (2020). †Results are from single run.

| MODEL | SIZE | STL-10 ACC. |
| --- | --- | --- |
| CIFARResNet-18 | 11.2M | 81.0\*† |
| FlexNet-16 | 0.67M | $68.6 \pm 0.7$ |

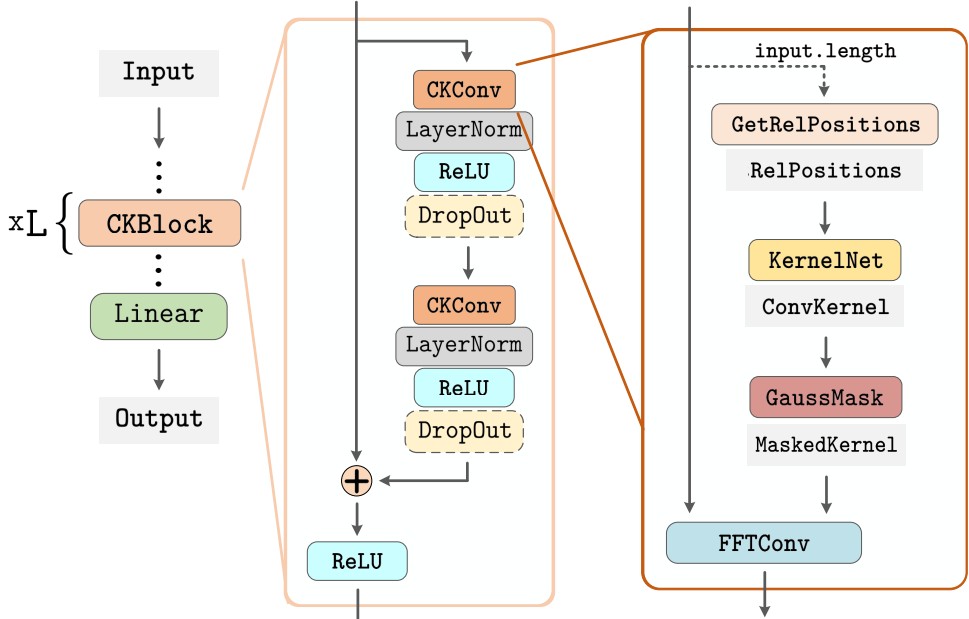

Figure 10: FlexNet architecture. FlexNet-L consists of L FlexBlocks, where each FlexBlock is a residual block of FlexConvs.

**FlexConv initialization.** We initialize the FlexConv mask variances small, at $\sigma_X^2, \sigma_Y^2 = 0.125$. For initializing MAGNet, we initialize the Gaussian envelopes as discussed in Sec. 3.2. We initialize the linear layer weights by the same Gamma distribution as used for the enveloped, modulated by a scaling factor of 25.6. We found that this value of the scaling factor, rather than a higher one, helped in reducing the performance of alias-free models. We initialize the bias of the linear layers by $\mathcal{U}(-\pi, \pi)$.

**CIFAR-10.** In FlexNet-16 models for CIFAR-10 we use $c = 24$ to approximate the parameter count of CIFARResNets in the experiment.

### D.2 OPTIMIZATION

We use Adam (Kingma & Ba, 2014) to optimize FlexNet. Unless otherwise specified, we use a learning rate of 0.01 with a cosine annealing scheme (Loshchilov & Hutter, 2016) with five warmup epochs. We use a different learning rate of $0.1\times$ the regular learning rate for the FlexConv Gaussian mask parameters. We do not use weight decay, unless otherwise specified.

**Kodak.** We overfit on each image of the dataset for 20,000 iterations. To this end, we use a learning rate of 0.01 without any learning rate scheme. We observe that SIRENs diverge with this learning rate and thus, reduce the learning rate to 0.001 for these models.

**CIFAR-10.** We train for 350 epochs with a batch size of 64. We use the data augmentation from He et al. (2016) when training CIFAR-10: a four pixel padding, followed by a random 32 pixel crop and a random horizontal flip.

**ImageNet-32.** We train for 350 epochs with a batch size of 2048. We use the same data augmentation as used for CIFAR-10. We do use a weight decay of $1e-5$ for ImageNet-32 training.

**Sequential and Permuted MNIST.** We train for 200 epochs with a batch size of 64 and a learning rate of 0.01. We use a weight decay of $1e-5$.

**Sequential and Noise-Padded CIFAR-10.** For sequential CIFAR-10, we train for 200 epochs with a batch size of 64, a learning rate of 0.001 and a weight decay of $1e-5$. For noise-padded CIFAR-10, we train for 300 epochs with a batch size of 32, a learning rate of 0.01 and no weight decay.

**Speech Commands and CharTrajectories.** We train for 300 epochs with a batch size of 32 and a learning rate of 0.001. For CharTrajectories, we use a weight decay of $1e-5$.

### D.3 ROTATED GAUSSIAN MASKS

MAGNets use anisotropic Gaussian terms in the Gabor filters, which yields improvements in descriptive power and convergence speed (Sec. 3.2). For the same reason, we explore making the anisotropic FlexConv Gaussian mask steerable, by including an additional vector of learnable angle parameters $\phi^{(l)} \in \mathbb{R}^{N_{\text{hid}}}$ that rotates the Gaussian masks. Although preliminary experiments show rotated masks lead to slight additional improvements, the computational overhead required to rotate the masks is large. Consequently, we do not consider rotated Gaussian masks in our final experiments.

