# OpenReview forum: "FlexConv: Continuous Kernel Convolutions With Differentiable Kernel Sizes"
_ICLR.cc/2022/Conference — ICLR 2022 Poster_

### Official Review · Reviewer_hUMj · 2021-10-31

**Correctness:** 3
**Technical Novelty And Significance:** 2
**Empirical Novelty And Significance:** 2
**Recommendation:** 6
**Confidence:** 3

**Main Review:**

Pros
The FlexConv is proposed to produce learn kernels with various kernel sizes.
An effective regularization is proposed to address the aliasing issue and generalization when dealing with higher resolution at the inference stage.

Cons
1. Incremental novelty compared to Multiplicative Filter Networks (Fathony et al., 2021). Anisotropic gabor functions are used to replace the original isotropic gabor functions in MFNs.
In addition, there is no explanation on why anisotropic gabor functions are used here and why it is better.
2. The idea of using gaussian mask to control the effective receptive field has also been proposed in Chapter 3.3 in [a]. The author even discussed different parametric kernel functions in that thesis.
[a] Duc Tam Nguyen, Robust Deep Learning for Computer Vision to counteract Data Scarcity and Label Noise, Doctoral Thesis, 2020.
3. There is no explanation on how the diversity in the initialization of $\gamma^{(l)}_X$ and $\gamma^{(l)}_Y$ relates to the accuracy.
4. There is no explanation on how equation 9 is derived.
5. Why not put relative change instead of absolute number in the left table of Figure 5, but use the absolute number in the right table?
6. Lack comparison with other methods on image benchmark datasets. Why only compare with ResNet44 instead of more popular ones such as ResNet50, ResNet 18, ResNet 101?
7. Lack comparison between a ResNet and the one whose all normal convolutional layers are replaced by FlexConv Layers.
8. The result of figure 6 is a little different from intuition and classic convolutional neural networks such as AlexNet and ResNet. Any explanation on this?

--------------------------------------------------------------

The explanation of authors has clarified some of my concerns but there are still some issues remained. Therefore I only raise my ratings to '6: marginally above the acceptance threshold'.

**Summary Of The Paper:**

This paper presents a novel convolutional operation named FlexConv, to produce high bandwidth convolutional kernels with learnable kernel size at a fixed parameter cost. It is able to generate kernels with large kernel size and model long-term dependencies among elements in a sequence or an image. State-of-the-art performance on both sequential datasets and image datasets demonstrates the effectiveness of the proposed method. In addition, a novel kernel parameterization method is proposed to control the frequency of the generated kernels and avoid aliasing, and hence can well generalize to higher resolution cases which are never seen during training.

**Summary Of The Review:**

The novelty of the paper is incremental and further explanation and experiment are required to demonstrate its effectiveness.

---

> ### Author Response · Authors · 2021-11-17
> **First response Reviewer hUMj**
>
> Dear reviewer hUMj,
>
> First of all, we would like to thank you very much for your thorough and insightful review. We sincerely appreciate the time you spend in evaluating our work, and very much appreciate your comments.
>
> Here we will answer to all of your questions, comments and concerns:
>
> > 1. Incremental novelty compared to Multiplicative Filter Networks (Fathony et al., 2021).
>
> Aside from the changes made to MFNs to obtain MAGNets, we are first in recognizing the value of MFNs to parameterize convolutional kernels. We believe this to be an important contribution as this not only allows us to get better classification results, but also allows us to control analytically the properties of the resulting approximation, e.g., for anti-aliasing: a property not easily attainable by other techniques such as SIRENs.
>
> > [T]here is no explanation on why anisotropic gabor functions are used here and why it is better. [...] 3. There is no explanation on how the diversity in the initialization of gamma_x and gamma_y relates to the accuracy.
>
> We provide two changes to MFNs to obtain MAGNets, namely (1) the change of isotropic Gaussian envelopes by anisotropic envelopes in order to construct anisotropic Gabor filters, and (2) an improved initialization scheme with which empirical gains are obtained.
>
> We agree our motivation for the anisotropic gaussians is brief and incomplete. We will provide a full motivation here, and complete it in the revision. We realize that isotropic envelopes do not allow basis functions to add frequencies in a certain direction without inducing it in a circle. Importantly, we are not the first to recognize this, but in fact, this is a fact known since the development of 2D Gabor filters. The seminal work that generalizes Gabor functions to 2D, argues the following regarding elliptical envelopes (Caption Fig 2 in [4]): “... Elongating the field instead in the perpendicular direction sharpens its spatial-frequency bandwidth but has no effect on its orientation bandwidth.” As a matter of fact, 2D Gabor functions are mainly modelled with an elliptical envelope in literature [1, 2, 4].
>
> With that being said, we indeed did not provide enough ablation studies to validate the individual contributions of MAGNet, both the initialization and the use of anisotropic envelopes. We show these results here and will include them in our revision.
>
> First, we follow [3], and use MFNs and MAGNets, both with and without our improved initialization, to approximate all of the images in the Kodak dataset (http://www.cs.albany.edu/~xypan/research/snr/Kodak.html), while using the same number of layers and parameters (10 layers, 23 hidden units). Our results empirically validate the significance of both the use of anisotropic Gaussian envelopes, and our improved initialization:
>
> | Model | PSNR (higher is better | $\Delta$ with MFN_Gabor |
> | --- | --- | --- |
> | MFN_Gabor | 25.390 | |
> | MFN_Gabor + anisotropic gaussians | 25.771 | +0.381 |
> | MFN_Gabor + improved init | 25.543 | +0.153 |
> | MAGNet | 25.884 | +0.494 |
>
> In addition, we conduct experiments on CIFAR-10 over three random seems for each of these combinations, which match the setting of the results in Tab. 3:
>
> - FlexNet-7 with MFN_Gabor: 91.9 +- 0.2
> - FlexNet-7 with MFN_Gabor + new initialization: 92.0 +- 0.2
> - FlexNet-7 with MFN_Gabor + anisotropic gaussians: 92.0 +- 0.1
> - FlexNet-7 with MAGNet: 92.2 +- 0.1
>
> Both these experiments show that both the individual contributions of MAGNets are beneficial. We will include our additional experiments and the extended motivation of anisotropic kernels in our revision of the paper.
>
> Note that in 1D the only difference between an MFN_Gabor and a MAGNet is the use of the improved initialization, because in 1D anisotropic and isotropic envelopes are equivalent. Our experimental results illustrate that this behavior also holds for these experiments.
>   .

---

> > ### Author Response · Authors · 2021-11-17
> > **First response Reviewer hUMj -- continuation ---**
> >
> > > 2. The idea of using a gaussian mask to control the effective receptive field has also been proposed in Chapter 3.3 in [a]. The author even discussed different parametric kernel functions in that thesis. [a] Duc Tam Nguyen, Robust Deep Learning for Computer Vision to counteract Data Scarcity and Label Noise, Doctoral Thesis, 2020.
> >
> > We regret to say we were not aware of this related work by Nguyen et al. [a]. Unfortunately, during our search while defining the name of our method, we did not come across this work. The chapter that discusses the related method has not been published elsewhere, which makes it hard to discover the FlexConv [a] method from the title of the thesis alone. The Google Scholar index for the work also does not provide a freely accessible PDF or any other metadata that refer to the relevant chapter, hindering the discovery of the chapter. Additionally, the work has not been cited by any of the works that were in our scope. We offer our most sincere apologies to the author of this work for using the exact same name for our method. This was not our intention.
> >
> > To compare our FlexConv against the FlexConv of [a], we observe that [a] does not learn a convolutional kernel, but rather an aggregation of pixels by weighting using $G$, which is then processed by further 1x1 convolutions. In our view, the difference with a convolutional kernel of learned size is that the method of [a] performs a sum over the learned window, which is equivalent to a low frequency kernel with a small bandwidth. In contrast, the kernels of our method can learn regular convolutional kernels with high bandwidth. This high bandwidth kernel is the core novelty of our work. We thank the reviewer for pointing this work out. We will make sure to cite [a] in our related works in the revision of our work.
> >
> > > 4. There is no explanation on how equation 9 is derived.
> >
> > This is true. We will include a formal derivation in a new appendix of our revision of the paper, and provide intuition here.
> >
> > Since our convolutional kernels are continuous, we can sample them at different resolutions. Consequently, if a certain input is presented in a down / upsampled way, we can down / upsample the convolutional kernel accordingly. As both the input and the convolutional kernel share the same resolution, the convolution will be equivalent, up to a factor proportional to the relative resolution change, which normalizes over the number of samples upon which the summation in the convolution is performed.
> >
> > > 5. Why not put relative change instead of absolute number in the left table of Figure 5, but use the absolute number in the right table?
> >
> > You are right. This would better illustrate the purpose of Fig. 5. We will change this in our revision.
> >
> > > 6. Lack comparison with other methods on image benchmark datasets. Why only compare with ResNet44 instead of more popular ones such as ResNet50, ResNet 18, ResNet 101?
> >
> > Note that the original ResNet paper [5] develops ResNets for two particular datasets: CIFAR-10 and ImageNet. In this paper, ResNet 18, 50, 101 are all ResNet versions developed for ImageNet. As illustrated in Figure 6, He et al. 2016 create ResNets 20, 32, 44, 56, 110 and 1202 specifically for CIFAR-10. Since these were the prime ResNet architectures developed for this dataset, we decided to use these as baselines, instead of the ResNets 18, 50, 101 developed for ImageNet.
> >
> > We understand this may not be obvious from the current revision of the work. We will clarify this in our revision by denoting the CIFAR-10 variants of  ResNets as “CIFARResNets”.
> >
> > > 7. Lack comparison between a ResNet and the one whose all normal convolutional layers are replaced by FlexConv Layers.
> >
> > We choose to show the effects of using convolutions in FlexNets, as shown in Tab. 3. We made this choice because we found that FlexConv do not need to be extremely deep in order to get high performance. Note that FlexNets are (shallow) CIFARResNets without pooling layers (Fig.  10). As such, we think it is interesting that FlexNet can be competitive with conventional CIFARResNets, without using subsampling and while being shallower.
> >
> > That said, even though FlexConv was not designed for use in a CIFARResNet, we provide the results for FlexConv/CKConv in CIFARResNet-44 below. We will include them in our revision. For CKConvs, we have replaced all convolutions with CKConvs of the same kernel size. For FlexConvs, we have replaced all convolutions with FlexConvs of learnable kernel size, except the shortcut connections, which remain 1x1 convolutions.
> >
> > CIFAR-10 accuracy:
> > CIFARResNet-44 (our reproduction): 90.9%
> > CIFARResNet-44 with CKConvs (k=3): 86.2%
> > CIFARResNet-44 with FlexConvs (learned kernel size, convolutions in shortcuts): 81.6%

---

> > > ### Author Response · Authors · 2021-11-17
> > > **First response Reviewer hUMj -- continuation --**
> > >
> > > > 8. The result of figure 6 is a little different from intuition and classic convolutional neural networks such as AlexNet and ResNet. Any explanation on this?
> > >
> > > In AlexNet-like networks [6], the kernel size does not change with depth. However, the receptive field size does, as pooling (AlexNet [6]) or striding (ResNet [5]) increases the receptive field of feature maps. In our FlexNets, which do not use pooling or striding, we find these same patterns, as shown in Fig. 6: filters in deeper layers learn to be larger in size than those in early layers, to cover the same receptive field as their AlexNet-like counterparts. This means FlexNets *learn* to use the same receptive field as is hardcoded for AlexNet-like networks.
> > >
> > > **Final words.** We hope that these responses clarify your questions and concerns. We will reflect this in an updated version of our manuscript by the end of this week. Please let us know if you have any follow-up / additional questions.
> > >
> > > Best regards,
> > >
> > > The Authors
> > >
> > > **References**
> > >
> > > [1] Movellan, J. R. (1996). Tutorial on Gabor Filters. https://inc.ucsd.edu/mplab/tutorials/gabor.pdf
> > >
> > > [2] Chao, W. (2010). Gabor wavelet transform and its application. http://disp.ee.ntu.edu.tw/~pujols/Gabor%20wavelet%20transform%20and%20its%20application.pdf
> > >
> > > [3] Dupont, E., Goliński, A., Alizadeh, M., Teh, Y. W., & Doucet, A. (2021). COIN: COmpression with Implicit Neural representations. arXiv preprint arXiv:2103.03123.
> > >
> > > [4] J. G. Daugman, "Complete discrete 2-D Gabor transforms by neural networks for image analysis and compression," in IEEE Transactions on Acoustics, Speech, and Signal Processing, vol. 36, no. 7, pp. 1169-1179, July 1988, doi: 10.1109/29.1644.
> > >
> > > [5] He, K., Zhang, X., Ren, S., & Sun, J. (2016). Deep residual learning for image recognition. In Proceedings of the IEEE conference on computer vision and pattern recognition (pp. 770-778).
> > >
> > > [6] Krizhevsky, A., Sutskever, I., & Hinton, G. E. (2012). Imagenet classification with deep convolutional neural networks. Advances in neural information processing systems, 25, 1097-1105.

---

### Official Review · Reviewer_T41g · 2021-11-01

**Correctness:** 2
**Technical Novelty And Significance:** 2
**Empirical Novelty And Significance:** 2
**Recommendation:** 6
**Confidence:** 3

**Main Review:**


Pros.
1.	This paper introduced a new parameterization method of CNN kernels by multiplying continuous kernels and with sigma-learnable Gaussian masks. Especially, multiplication with learnable Gaussian is first demonstrated.
2.	Flexconv operation is robust to fit high frequency signals in large kernels by MagNets.
3.	According to Table 4, the proposed network shows good performance on higher resolution images without additional training.
4.	This work achieves state-of-the-art performance on several sequential datasets.

Cons.
1.	“Large kernels” and “small kernels” are too obscure.
2.	Although the meaning of “bandwidth” seems to be important, the descriptions are not enough to understand. It confuses the original meaning of bandwidth; originally, bandwidth in signal processing is a term for finite support in frequency domain.
3.	In table 4, the two scales of 16 pixels and 32 pixels are inefficient to explain the performance of FlexConv. This is exaggerated without further scale experiments.
4.	The authors claimed that the cropping operation in Flexconv improves computational efficiency. But there is no ablation study about the cropping operations, reviewer has no clues about the relationship between cropping size and computational efficiency.

Minor.

In page 20, there are type such as ‘??’.


**Summary Of The Paper:**

This paper proposed a new convolutional operation with varying-sized kernels. Flexconv constructs their convolutional kernels with the product between the CKConv and Gaussian masks so that the size of the kernels can be learned during training. Also, in the network of Flexconv, MAGNets allow the Flexconv to be used at higher resolutions without aliasing which are unseen resolution. The experiment on several sequential datasets showed that the classification performances from both TCN and ResNet based on FlexConv outperform that of state-of-the-art networks.

**Summary Of The Review:**

The way of representing and analyzing the results is a bit overclaimed without serious supporting experiments. However, their method demonstrated state-of-the-art performances compared with recent approaches.

---

> ### Author Response · Authors · 2021-11-17
> **First response Reviewer T41g**
>
> Dear reviewer T41g,
>
> First of all, we would like to thank you very much for your thorough and insightful review. We sincerely appreciate the time you spend in evaluating our work, and very much appreciate your comments. We appreciate that you recognize our contributions as well as the significant empirical results achieved by our method.
>
> Here we will answer to all of your questions, comments and concerns:
>
> > 1. “Large kernels” and “small kernels” are too obscure.
>
> The definitions of “large” and “small” kernels are indeed vague, especially since we consider both the sequence and image classification domains, where these definitions may conflict.
>
> To clarify, mall kernels are considered to be of size in the order of 3, 5 or 7, which are the kernel sizes used in seminal works like VGG [1], Inception [2] and ResNet [3] for images, and TCN [4] for sequences. We will consider kernels larger than this to be “large” kernels. We will clarify this in our revision.
>
> > 2. Although the meaning of “bandwidth” seems to be important, the descriptions are not enough to understand. It confuses the original meaning of bandwidth; originally, bandwidth in signal processing is a term for finite support in the frequency domain.
>
> To the best of our understanding, we do use the same meaning of bandwidth in our work.
> In our understanding, the bandwidth of a kernel is the finite support of the kernel in the frequency domain. The frequency response of our kernels has a larger support than those of the related works, as shown in the experiment in Sec. 4.1, where we show that MAGNet can respond to a wider range of Gabor filter frequencies than N-Jets (see the left plot of Fig. 4).
>
> > 3. In table 4, the two scales of 16 pixels and 32 pixels are inefficient to explain the performance of FlexConv. This is exaggerated without further scale experiments.
>
> We would agree that merely testing FlexConv on translating between 16px and 32px would be insufficient. Note, however, that Fig. 5 shows results for ten different scale combinations. This confirms that the translated performance is consistent for many scales.
> We will change the reference to Fig. 5 to more explicitly refer to the comparisons across different scales.
>
> > 4. The authors claimed that the cropping operation in Flexconv improves computational efficiency. But there is no ablation study about the cropping operations, reviewer has no clues about the relationship between cropping size and computational efficiency.
>
> The computational cost of performing a convolution between a signal of size M and a kernel of size k is O(Mk). Naturally, the smaller the size of the kernel $k$, the lower the computational complexity of the convolution will be. This effect is exacerbated in 2D, where this complexity becomes O(NMk_1 k_2) for an image of size MxN and a kernel of size k_1 x k_2 . If we now crop the size of the kernel in these two dimensions, the computational complexity of the operation is reduced at a faster rate.
>
> In practice, the question remains whether cropping is faster than simply taking the large kernel. To test this we take a signal from the Speech Commands dataset, and convolve it with a kernel of size 16000 and a kernel of size 1600 obtained via dynamic cropping. The running times are:
>
> |No cropping | Cropping |
> | --- | --- |
> |  41.2609     |    3.5153  |
>
> This shows that indeed cropping can strongly reduce the complexity of the operation ( 11.73x in this example). However, we agree that this property was not accentuated enough in the paper. To make it clear, we will add the previous analysis to our paper, and include a small paragraph in the discussion section.

---

> > ### Author Response · Authors · 2021-11-17
> > **First response Reviewer T41g -- continuation --**
> >
> > > In page 20, there are type such as ‘??’.
> >
> > Thank you for pointing this out. We will fix these typos in our revision.
> >
> > > [Summary of the review]: The way of representing and analyzing the results is a bit overclaimed without serious supporting experiments. However, their method demonstrated state-of-the-art performances compared with recent approaches.
> >
> > We hope that our responses alleviate the reviewer's concerns regarding overclaiming of our empirical results. If there is anything else we can do to strengthen our paper’s positioning in this respect, please let us know.
> >
> > **Final words.** We hope that these responses clarify your questions and concerns. We will reflect this in an updated version of our manuscript by the end of this week. Please let us know if you have any follow-up / additional questions.
> >
> > Best regards,
> >
> > The Authors
> >
> > **References**
> >
> > [1] Simonyan, K., & Zisserman, A. (2014). Very deep convolutional networks for large-scale image recognition. arXiv preprint arXiv:1409.1556.
> >
> > [2] Szegedy, C., Liu, W., Jia, Y., Sermanet, P., Reed, S., Anguelov, D., ... & Rabinovich, A. (2015). Going deeper with convolutions. In Proceedings of the IEEE conference on computer vision and pattern recognition (pp. 1-9).
> >
> > [3] He, K., Zhang, X., Ren, S., & Sun, J. (2016). Deep residual learning for image recognition. In Proceedings of the IEEE conference on computer vision and pattern recognition (pp. 770-778).
> >
> > [4] Bai, S., Kolter, J. Z., & Koltun, V. (2018). An empirical evaluation of generic convolutional and recurrent networks for sequence modeling. arXiv preprint arXiv:1803.01271.

---

> > > ### Author Response · Authors · 2021-11-22
> > > **Correction to rebuttal to reviewer T41g**
> > >
> > > In our rebuttal on November 17, we claimed the following:
> > >
> > > > This shows that indeed cropping can strongly reduce the complexity of the operation ( 11.73x in this example).
> > >
> > > In the process of revising the paper, we have found that cropping does not always yield benefits over not cropping the FlexConv kernels. The benefits of cropping are undone by PyTorch’s optimization algorithms. Without these optimizations, cropping does improve the computational efficiency of FlexConv.
> > >
> > > Please see the comment we posted on our revision for full details.

---

### Official Review · Reviewer_APZa · 2021-11-01

**Correctness:** 3
**Technical Novelty And Significance:** 3
**Empirical Novelty And Significance:** 3
**Recommendation:** 6
**Confidence:** 4

**Main Review:**


(1) Section 4.1 is not very clear, how the target image is fitted by different kernels? What is an AlexNet kernel, 3x3, 7x7? Please give more descriptions of the experiment.

(2) In Table 3, it would be better to show the time consumption of ResNet-44 or its reproduction since FlexNet has additional operations that will reduce the inference speed, which is much more crucial than the model size.

(3)  This paper and its related work focus on kernel sizes, dilations, etc., which may obtain greater improvement on high-resolution images. However, it is not clear whether FlexNet performs well on high-resolution image tasks, such as ImageNet or MS-coco in the cited work [1].
The authors are expected to demonstrate the superiority of FlexNet on typical and practical tasks (classification, generalization, detection), which would draw more attention from the CNN community.

(4) Why "alias-free" is important? The authors claim that FlexNets can be deployed at higher resolutions than those seen during training. But the experiments only show the improvement on accuracy. It would be more clear if authors can show the superiority on other aspects, for example, vanilla nets are trained on high resolutions while FlexNet are trained on lower resolutions, they have comparable classification accuracy but the latter are more efficient in terms of time/computation cost.

[1] Dai, Jifeng, et al. "Deformable convolutional networks." Proceedings of the IEEE international conference on computer vision. 2017.

==========================================================================

I have read the authors' rebuttal, some of the concerns have been addressed, others still remain.
The authors show that the reproduction of the ResNet-44 is 22s per epoch during training, which is much faster than FlexNet series with similar # parameters and accuracy, which almost can be extended to the inference stage.
In the reply to point 4, the author also notes the computational efficiency of alias-free models.
However, a direct comparison between FlexNets and classical models (ResNet, DenseNet, etc) with regard to the accuracy, training time, model size, inference time, etc, is still lacking.
Aside from ablation study in the settings of FlexNet models, the authors are encouraged to provide more evidence such that the community is willing to discard common practices and adopt your methods on some occasions.
In summary, the reviewer will keep the score.



**Summary Of The Paper:**

The authors propose FlexConv, a novel convolutional operation with which high bandwidth convolutional kernels of learnable kernel
size can be learned at a fixed parameter cost. The FlexNet obtains state-of-the-art across several sequential datasets, and matchs recent works with learnable kernel size on CIFAR-10 with less compute.

**Summary Of The Review:**

This is a very interesting study, the method and theoretical analysis seem sold, but the motivation and comparative experiments should be improved.

---

> ### Author Response · Authors · 2021-11-17
> **First response Reviewer APZa**
>
> Dear reviewer APZa,
>
> First of all, we would like to thank you very much for your thorough and insightful review. We sincerely appreciate the time you spend in evaluating our work, and very much appreciate your comments. We appreciate that you perceive our paper as solid and interesting.
>
> Here we will answer to all of your questions, comments and concerns:
>
> > (1) Section 4.1 is not very clear, how the target image is fitted by different kernels? What is an AlexNet kernel, 3x3, 7x7? Please give more descriptions of the experiment.
>
> We agree the setup of this experiment is not clear in the paper. The code for this experiment is provided in the supplementary material, and we will add the specific details to the paper:
>
> Each target function is fitted by an MLP by optimizing the pixel-wise mean squared error using Adam with a learning rate of 0.01, until convergence. In the case of the AlexNet kernel, we have taken a random 11x11 kernel from the first layer of the torchvision AlexNet pretrained model [1].
>
> > (2) In Table 3, it would be better to show the time consumption of ResNet-44 or its reproduction since FlexNet has additional operations that will reduce the inference speed, which is much more crucial than the model size.
>
> We have timed our reproduction of the ResNet-44 at 22s per epoch. We will add this information in our revision.
>
> > (3) This paper and its related work focus on kernel sizes, dilations, etc., which may obtain greater improvement on high-resolution images. However, it is not clear whether FlexNet performs well on high-resolution image tasks, such as ImageNet or MS-coco in the cited work [1]. The authors are expected to demonstrate the superiority of FlexNet on typical and practical tasks (classification, generalization, detection), which would draw more attention from the CNN community.
>
> FlexConv provides important improvements in the efficiency of CKConv. However, this is not enough to run experiments on image datasets like the full-resolution ImageNet with our resources. Note, however, that our kernel fitting experiments (Sec. 4.1) indicate the MAGNets can fit high-resolution signals In addition, we show that FlexNets obtain significant improvements in high-resolution sequential data (Sec. 4.2). For instance, FlexNets achieve state of the art in Speech Commands raw, whose sequence length is 16000.
>
> > (4) Why "alias-free" is important? The authors claim that FlexNets can be deployed at higher resolutions than those seen during training. But the experiments only show the improvement on accuracy. It would be more clear if authors can show the superiority on other aspects, for example, vanilla nets are trained on high resolutions while FlexNet are trained on lower resolutions, they have comparable classification accuracy but the latter are more efficient in terms of time/computation cost.
>
> Excellent point. Alias-free kernels allow performance of FlexNets to generalize across resolutions, which is a feature not exhibited by conventional CNNs. However, in the current revision, we indeed do not give a clear argument for the benefits of generalization across resolutions.
>
> Our main motivation is that training at a low resolution is more computationally efficient than at a high resolution, as the complexity of convolution depends on the image size. With a smaller image size the complexity of the convolutional operation is decreased, reducing inference times.
>
> In practice, this difference holds true. We time the first 32 batches of training a FlexNet-7 on CIFAR-10, where we compare training on 16x16 images (downsampled before training) against training on 32x32 images. On 16x16 images, each batch takes 179ms (+- 7ms). On 32x32 images, each batch takes 222ms (+- 9ms).
>
> We will include this motivation and these results in our revision of the paper.
>
> **Final words.** We hope that these responses clarify your questions and concerns. We will reflect this in an updated version of our manuscript by the end of this week. Please let us know if you have any follow-up / additional questions.
>
> Best regards,
>
> The Authors
>
> **References**
>
> [1] https://pytorch.org/vision/stable/models.html#id1

---

> > ### Author Response · Authors · 2021-11-22
> > **Correction rebuttal to reviewer APZa**
> >
> > In our rebuttal on November 17, we claimed the following:
> >
> > > FlexConv provides important improvements in the efficiency of CKConv.
> >
> > In the process of revising the paper, we have found that the reported timing of CKConv is incorrect. CKConv is in fact faster than FlexConv. We discuss this finding extensively in the official comment posted along with our revision. In short, FlexConv uses the more expensive MFN instead of SIREN, and the benefits of cropping are undone by PyTorch’s optimization algorithms. Without these optimizations, FlexConv is still faster than CKConv.
> >
> > Please see the comment to our revision for full details.

---

### Official Review · Reviewer_ub8r · 2021-11-02

**Correctness:** 3
**Technical Novelty And Significance:** 3
**Empirical Novelty And Significance:** 2
**Recommendation:** 8
**Confidence:** 4

**Main Review:**

### Strengths
1. Nice ideas and well-argumented design
2. A mechanism to control aliasing
3. Results above those of existing works

__Idea.__ The approach is based on the idea of allowing the learning of filters of different size within a convolutional network, and it is supported by literature and empirical studies. The use of continuous convolutions via implicit representations is becoming of interest for the community. This work has thus relevance and places itself well within the current developments in representation learning.

__Anti-aliasing.__ The observation that other implicit representations (e.g. SIREN) suffer from aliasing when kernels learned for a given resolution are applied to higher resolution data stimulated the authors to investigate the possibility of learning an anti-aliased implicit representation network. The use of anisotropic Gabor functions as kernels for the implicit representation networks (which can be considered, in my opinion, the true contribution of this work) allows for controlling the maximum learnable frequency.

__Results.__ The results obtained are higher than those reported by other existing works. This places this paper as state-of-the-art for learnable kernel networks.


### Weaknesses
1. Contributions are not made crystal clear. Novelty seems oversold.
2. The work seems a marginal improvement on CKConv, with the difference that an anisotropic Gabor kernel is used to modulate the implicit network representations.
3. The computational impact of the learnable kernels is not discussed enough. Application to 'normal-size' images is
4. Some comparison statements are questionable.

__Contributions.__ In my opinion, the authors make overstatements about their contributions. FlexConvs seems like a re-branding of CKConvs, while FlexNets are networks that deploy FlexConvs but are presented as a contribution themselves. The true contribution of this paper, and the main part where novelty is presented, can be considered the use of anisotropic Gabor functions as kernels of the implicit representation networks. The Gabor functions indeed allow to control the extension of the Gaussian masks that determine the size of the kernels, and also constrain the maximum learnable frequency for the kernels in order to avoid aliasing in the networks.

__CKConv.__ This work seems to build a lot on top of CKConvs (also under review @ICRL22 - here results of CKConv are reported, not likewise for the CKConv paper), but fails to highlight what the real contribution is (see point above).

__Computational impact.__ The computational impact of using MAGNets is only briefly addressed in this paper. An obvious observation is that bigger kernels require more computations because of the increased number of convolutional weights. This also relates to the increased size of the images in the case a network is applied to higher-resolution images (the case of a network learned on CIFAR 16x16 and then upscaled to CIFAR 32x32). In this work the maximum threated resolution is indeed 32x32, also in the case of ImageNet, which is rescaled. An actual discussion and experiments about the learning or application (by upscaling) of the proposed approach to REAL higher-resolution images (e.g. the original ImageNet) are missing. This limits, in my opinion, the transferrability of this work to real-case uses.

__Questionable statements.__ Some statements about results are questionable or not precise. For instance, the authors claim that the proposed FlexNet achieves comparable results to much deeper ResNets. Actually, the ResNet and the FlexNet concerned by this statement have roughly the same number of parameters, which may influence more directly the learning capabilities of the networks (and thus the results). The ResNet has  blocks (although more convolutional layers), while the FlexNet has 7 FlexConv Blocks. Should it then be considered deeper than the ResNet? I think that the authors should not bring the depth as an argument at their advantage, but rather focus on comparing the networks in terms of the learnable parameters.


On a general note, the paper introduces nice ideas, but it should be better written to highlight the contributions and what is taken from the state-of-the-art.

=========================================================================
After reading the response to reviews and the revision of the paper, I had my concerns clarified. I identify the novelty and amount of contributions provided by the authors in this paper, and recognize their value for further development of CNNs. Although relation with training at low resolution and testing at higher resolution is 'only' drafted in this paper, one can deal with the fact that this paper is a good seed for further experiments in future works. Thus I am inclined to raise my score.



**Summary Of The Paper:**

The paper proposes a method for learning convolutional filters with trainable size, that builds on top of multiplicative filter networks. The learnable convolutions are called FlexConv, and a network that deploys them is called FlexNet. In order to control the aliasing eventually introduced when learning the convolutional filters, anisotropic Gabor kernels are used within the multiplicative filter networks.

**Summary Of The Review:**

The paper introduces some new nice idea, but it seems overstating some of the contributions. The experimental analysis is extensive but on very small data set of low-resolution, while actual problems encountered to scale this method up to more 'real-case' resolutions are not discussed appropriately.

---

> ### Author Response · Authors · 2021-11-17
> **First response Reviewer ub8r**
>
> Dear reviewer ub8r,
>
> First of all, we would like to thank you very much for your thorough and insightful review. We sincerely appreciate the time you spend in evaluating our work, and very much appreciate your comments. We are glad you appreciate the ideas of the paper and its relevance to the representation learning community.
>
> Here we will answer to all of your questions, comments and concerns:
>
> > In my opinion, the authors make overstatements about their contributions. FlexConvs seems like a re-branding of CKConvs [...]
>
> We believe the addition of the gaussian mask to CKConv [1] is an important contribution to improve CKConvs. It brings two major improvements over CKConvs:
>
> 1) The addition of the gaussian mask to CKConv enables FlexConv to learn its kernel size. CKConvs are not able to learn their kernel size. They must use a global kernel size and are therefore quite inefficient for usage on images: where CKConv takes 266s to train one epoch on CIFAR-10, FlexConv takes 166s (see Tab. 3). In Fig. 6 we show that FlexConv learns much smaller kernels than CKConv, confirming the use of learnable kernel size.
>
> 2) Crucially, the gaussian mask allows FlexConv to focus its bandwidth on a smaller kernel (see Fig. 3). This leads to significant improvements (92.2%) over the baseline CKConv with MAGNet (85.9%) on CIFAR-10 (see Tab. 3).
>
> > [...] while FlexNets are networks that deploy FlexConvs but are presented as a contribution themselves.
>
> From our introduction, it seems we indeed present FlexNet as a significant contribution on its own. Instead, we intended for FlexNet merely to be a vessel for the experimental validation of FlexConv, while remarking on its properties as interesting findings. It seems that by giving the network a specific name we may have implied it is more significant than it is as a contribution. We will de-emphasize FlexNet as a separate contribution in the introduction in our revision, though we retain the name “FlexNet”, as it has a role in distinguishing the networks from ResNets in our experiment.
>
> > *CKConv.* This work seems to build a lot on top of CKConvs (also under review @ICRL22 - here results of CKConv are reported, not likewise for the CKConv paper), but fails to highlight what the real contribution is (see point above).** We will make sure to make clear what the contributions of our work are. Aside from the contributions related to MAGNets, we are, to the best of our knowledge, the first work that presents a way to learn convolutional kernels of changing kernel sizes which mantain high-bandwidth.
>
> We replied to this concern in our reply to the first comment.
>
> > The computational impact of using MAGNets is only briefly addressed in this paper. An obvious observation is that bigger kernels require more computations because of the increased number of convolutional weights.
>
> Larger kernels indeed are more computationally expensive. We show that CKConv [1], which uses kernels of the size of the image, is quite inefficient on CIFAR-10 images (266s per epoch, see Tab. 3). Our contribution of the gaussian mask in FlexConv lessens this computational burden by learning the kernel size of CKConv, which can learn to use smaller kernels. Indeed, Fig. 6 shows FlexConvs learn smaller kernels than the global kernels of CKConv, thereby improving the computational efficiency of CKConv.
>
> > [...] REAL higher-resolution images (e.g. the original ImageNet) are missing. This limits, in my opinion, the transferability of this work to real-case uses.
>
> Although it is true that we do not perform experiments on high-resolution images, we do so for high-resolution sequences. Note that FlexConvs achieve state-of-the-art on SpeechCommands: a high-resolution audio dataset with sequences of length 16000.
>
> For high-resolution images, we consider FlexConv an important improvement to the computational efficiency of CKConv, as a step in the direction of making continuous kernel convolutions work efficiently on high-resolution images.

---

> > ### Author Response · Authors · 2021-11-17
> > **First response Reviewer ub8r -- continuation --**
> >
> > > Some statements about results are questionable or not precise. For instance, the authors claim that the proposed FlexNet achieves comparable results to much deeper ResNets. Actually, the ResNet and the FlexNet concerned by this statement have roughly the same number of parameters, which may influence more directly the learning capabilities of the networks (and thus the results). The ResNet[-44] has [21] blocks (although more convolutional layers), while the FlexNet has 7 FlexConv Blocks. Should it then be considered deeper than the ResNet? I think that the authors should not bring the depth as an argument at their advantage, but rather focus on comparing the networks in terms of the learnable parameters.
> >
> > It seems our writing may have caused some naming confusion about the depths of ResNets and FlexNets: a FlexNet-x has x blocks, while a ResNet-x has x layers. In the revision, we will rename our FlexNets to denote the amount of layers. To clarify this point, however, A FlexNet-7 has 16 FlexConv layers, which is significantly less than the 44 convolutional layers of a ResNet-44.
> >
> > FlexNets and ResNets indeed have roughly the same amount of parameters in our experiments. We intentionally design FlexNets in this way to keep the number of parameters of the network (and thus their capacity) equal regardless of depth. Note that this allows us to disentangle the contributions from depth from the contributions resulting from the capacity of the network. If we were to construct FlexNets with fewer parameters, it would be difficult to study the influence of depth alone in the performance of the model.
> >
> > The lesser depth of FlexNets is featured as a finding since it seems to contradict a popular paradigm in Deep Learning, most notably discussed by He et al. [2], namely that depth is beneficial to scale performance of neural networks. The Wide ResNet [3] work, as well as another recent work [4] show that shallow but wide networks can have benefits over deep networks, as they are more easily parallelizable.
> >
> > **Final words.** We hope that these responses clarify your questions and concerns. We will reflect this in an updated version of our manuscript by the end of this week. Please let us know if you have any follow-up / additional questions.
> >
> > Best regards,
> >
> > The Authors
> >
> > **References**
> >
> > [1] Romero, D. W., Kuzina, A., Bekkers, E. J., Tomczak, J. M., & Hoogendoorn, M. (2021). CKConv: Continuous Kernel Convolution For Sequential Data. arXiv preprint arXiv:2102.02611.
> >
> > [2] He, K., Zhang, X., Ren, S., & Sun, J. (2016). Deep residual learning for image recognition. In Proceedings of the IEEE conference on computer vision and pattern recognition (pp. 770-778).
> >
> > [3] Zagoruyko, S., & Komodakis, N. (2016). Wide residual networks. arXiv preprint arXiv:1605.07146.
> >
> > [4] Anonymous. (2022). Non-deep Networks. Submitted to The Tenth International Conference on Learning Representations. Retrieved from https://openreview.net/forum?id=Xg47v73CDaj

---

> > > ### Author Response · Authors · 2021-11-22
> > > **Correction to rebuttal to reviewer ub8r**
> > >
> > > In our rebuttal on November 17, we claimed the following:
> > >
> > > > 1. The addition of the gaussian mask to CKConv enables FlexConv to learn its kernel size. CKConvs are not able to learn their kernel size. They must use a global kernel size and are therefore quite inefficient for usage on images: where CKConv takes 266s to train one epoch on CIFAR-10, FlexConv takes 166s (see Tab. 3). In Fig. 6 we show that FlexConv learns much smaller kernels than CKConv, confirming the use of learnable kernel size.
> > >
> > > and
> > >
> > > > We show that CKConv [1], which uses kernels of the size of the image, is quite inefficient on CIFAR-10 images (266s per epoch, see Tab. 3). Our contribution of the gaussian mask in FlexConv lessens this computational burden by learning the kernel size of CKConv, which can learn to use smaller kernels. Indeed, Fig. 6 shows FlexConvs learn smaller kernels than the global kernels of CKConv, thereby improving the computational efficiency of CKConv.
> > >
> > > In the process of revising the paper, we have found that the reported timing of CKConv is incorrect. CKConv is in fact faster than FlexConv. We discuss this finding extensively in the official comment posted along with our revision. In short, FlexConv uses the more expensive MFN instead of SIREN, and the benefits of cropping are undone by PyTorch’s optimization algorithms. Without these optimizations, FlexConv is still faster than CKConv.
> > >
> > > Please see our comment for full details.

---

### Author Response · Authors · 2021-11-22
**Comment on November 22th revision**

We have revised our paper after the comments of the reviewers. Along with the requested changes, we have made some other changes, which we detail below.

**Changes after reviewer comments.** We have made the changes we have mentioned in the previously submitted rebuttals. Most notably, we have extended our ablation studies on MAGNets for CIFAR-10, and added an image fitting experiment on the Kodak dataset as an additional ablation study. We have added ResNet-based baselines on CIFAR-10 to Table 6 in the appendix. We also now explicitly discuss the compute benefits of alias-free training at the end of Sec. 4.3.

**Other changes.** In the appendix, we have added more details on the exact regularization strategy followed in our anti-aliasing regularization. We have also improved the wording on the dataset and network specifications in the appendices.

**Inaccuracies.** In the process of revision, we discovered two inaccuracies in our original work:

*1. Efficiency of CKCNN.* We discovered that the time we reported for CKCNN in Tab. 3 was incorrect. As part of the new ablation studies we have run at the request of the reviewers, we have also corrected the timings of the ablations. Importantly, we have found that CKCNN is not slower, but faster than FlexConv.

Our ablation study shows that this is due to MFNs being significantly slower than SIRENs. Furthermore, the theoretical efficiency of the kernel size cropping that FlexConv enables does not come true in practice, as it seems the computational overhead of the cropping operation negates the benefits of computing a convolution with a smaller kernel. We have added this finding in the Discussion.

We have changed the revision to reflect these discoveries, and have replied with corrections to our original rebuttals to the reviewers who raised discussion about the computational efficiency of FlexConv/CKConv. In particular, we have removed claims that FlexConv is more computationally efficient than CKConv.

*2. Figure 5.* Some values in the “final test accuracy” of $f^+_{\textrm{FlexConv}}$ were permuted. We have arranged them correctly in the revision.

---

### Decision · Program_Chairs · 2022-01-20

**Decision:**

Accept (Poster)

**Comment:**

This submission proposes a method for learning convolutional filters with trainable size, that builds on top of multiplicative filter networks.  Anti-aliasing is achieved by parametrization with anisotropic Gabor filters.  The reviewers were unanimous in their opinion that the paper is suitable for acceptance to ICLR.  The authors are encouraged to make use of the extensive reviewer discussion in improving the final version of the paper.